# Sub-Doppler optical-optical double-resonance spectroscopy using a cavity-enhanced frequency comb probe

Vinicius Silva de Oliveira [1], Isak Silander [1], Lucile Rutkowski[2], Grzegorz Soboń [3], Ove Axner [1], Kevin K. Lehmann [4] & Aleksandra Foltynowicz [1] ✉

Accurate parameters of molecular hot-band transitions, i.e., those starting from vibrationally excited levels, are needed to accurately model high-temperature spectra in astrophysics and combustion, yet laboratory spectra measured at high temperatures are often unresolved and difficult to assign. Optical-optical double-resonance (OODR) spectroscopy allows the measurement and assignment of individual hot-band transitions from selectively pumped energy levels without the need to heat the sample. However, previous demonstrations lacked either sufficient resolution, spectral coverage, absorption sensitivity, or frequency accuracy. Here we demonstrate OODR spectroscopy using a cavity-enhanced frequency comb probe that combines all these advantages. We detect and assign sub-Doppler transitions in the spectral range of the $3v_3 \leftarrow v_3$ resonance of methane with frequency precision and sensitivity more than an order of magnitude better than before. This technique will provide high-accuracy data about excited states of a wide range of molecules that is urgently needed for theoretical modeling of high-temperature data and cannot be obtained using other methods.

Absorption spectroscopy is one of a few techniques that allow in-situ analysis of high-temperature molecular gases, with applications including combustion science[1–3] and atmospheric sensing of hot astrophysical objects[4,5]. Identification of the species and their densities from absorption measurements requires accurate theoretical models of the molecular ro-vibrational energy structure that are verified using high-precision laboratory spectra. The launch of the James Webb Space Telescope (JWST) has placed in the spotlight the immediate need for rotationally resolved reference spectroscopic data for many small organic species in a wide range of thermodynamic conditions[5]. The first JWST survey of the VHS-1256b exoplanet[6] yielded observational evidence of atmospheric $H_2O$, CO, $CO_2$, and $CH_4$. These unambiguous detections were based on a comparison of the detected spectral features to synthetic spectra computed at the relevant temperature (1000 K). However, these models do not account for all observed transitions and many features remained unidentified. Recently, $CH_3^+$, a radical cation related to methane photochemistry, has been identified in a hot protoplanetary disk[7] by comparing its emission spectrum recorded by the JWST to a model spectrum developed based on available spectroscopic constants. The remaining discrepancies between the model and the spectrum imply either the presence of other species or inaccuracies in the model[7]. Resolving this question requires verification of the theoretical model using experimental data, which is not available.

Such unresolved detections at high temperatures underline the urgent need for precision measurements of molecular excited states to provide the spectroscopic parameters required to model the hot bands. Among the different species, methane poses a particular

[1]Department of Physics, Umeå University, 901 87 Umeå, Sweden. [2]University of Rennes, CNRS, IPR (Institut de Physique de Rennes)-UMR 6251, F-35000 Rennes, France. [3]Faculty of Electronics, Photonics and Microsystems, Wrocław University of Science and Technology, Wybrzeże Wyspiańskiego 27, 50-370 Wrocław, Poland. [4]Departments of Chemistry & Physics, University of Virginia, Charlottesville, VA 22904, USA. ✉e-mail: aleksandra.foltynowicz@umu.se

challenge. The fundamental vibrational mode frequencies of methane are nearly resonant and coupled through a number of strong interactions. The high density of excited levels and the strong couplings between them make them difficult to calculate using ab-initio methods[8]. State-of-the-art synthetic high-temperature line lists of methane are contained in, e.g., the TheoReTS[9] and the ExoMol databases[10]. The TheoReTS data have recently been incorporated into the HITEMP database[11], which is used as a reference in many high-temperature applications. However, the energy levels above 8000 cm$^{-1}$—relevant to hot environments (>500 K)—remain largely unverified. Room- and low-temperature precision spectroscopy of overtone bands provides valuable information about levels that can be reached from the ground vibrational level[12] but often does not shed light on levels involved in hot-band transitions (i.e., transitions starting from excited vibrational states). Obtaining empirical hot-band line lists from laboratory measurements is difficult because absorption and emission spectra measured at high temperatures are often congested with overlapping transitions, making them difficult to resolve and assign[13–15].

Optical–optical double-resonance (OODR) spectroscopy is a method that allows selective measurement of hot-band transitions without the need to heat the sample. In OODR, a strong pump laser populates a selected excited state, and a weaker probe laser measures hot-band transitions from this state, which results in a much less congested spectrum, simpler to analyze than a spectrum from a thermally excited sample. OODR spectroscopy has historically been performed using tunable pulsed lasers[16] that provide broad spectral coverage but have limited spectral resolution and frequency accuracy, which often prevents resolving individual transitions. Compared to that, OODR spectroscopy using narrow-linewidth continuous-wave (CW) lasers has a number of advantages[17,18]: sub-Doppler resolution, because a pump with narrow linewidth excites only one velocity group of molecules; high absorption sensitivity, especially when combined with cavity-enhanced methods; and kHz frequency accuracy when the pump and probe lasers are referenced to a frequency comb. However, the tunability of narrow-linewidth CW lasers is limited, and surveying large spectral ranges is time-consuming, often making the search for transitions impractical and cumbersome.

Using frequency combs as probes in OODR spectroscopy overcomes these limitations and provides spectra with broad bandwidth, inherent absolute frequency calibration, and sub-Doppler resolution, allowing unambiguous detection of many hot-band transitions simultaneously. OODR based on a CW pump and a frequency comb probe was first performed on an atomic Rb sample[19,20] contained in a single-pass cell. This was possible because atomic transitions are 3–4 orders of magnitude stronger than molecular ones. Recently, we demonstrated OODR spectroscopy on a molecular sample[21] and used it to measure and assign 36 sub-Doppler transitions in the $3\nu_3 \leftarrow \nu_3$ resonance region of methane in probe spectra spanning 6 THz of bandwidth[22]. These results provided the first high-accuracy verification of theoretical predictions of hot-band transitions of methane in this range, finding better agreement with TheoReTS than ExoMol. However, the absorption sensitivity and frequency accuracy of these measurements were limited by the use of a single-pass cell for the methane sample. Even though the cell was cooled by liquid nitrogen to increase the intensity of the OODR probe transitions (by increasing the molecular population in the pumped low rotational states), the signal-to-noise ratio (SNR) of the OODR signal was at most 10. The requirement of cooling limited the applicability of the technique to methane, which is the only stable polyatomic molecule that has sufficient vapor pressure at 77 K. Moreover, in the cell, the pump and probe beams were co-propagating, and thus interacting with the same velocity group of molecules, so a residual drift of the pump laser frequency translated to a proportional shift of the probe transition frequencies, which limited the frequency accuracy.

Here, we introduce OODR spectroscopy using a cavity-enhanced frequency comb as the probe, which dramatically increases the absorption sensitivity and frequency precision of detection of hot-band transitions without the requirement of cooling of the sample, making it applicable to a large range of molecules. The cavity increases the interaction length of the probe with the sample, which allows the detection of transitions that are more than an order of magnitude weaker than previously observed. Moreover, while the pump beam makes a single pass through the sample, the cavity-enhanced probe beam is both co- and counter-propagating with respect to the pump and simultaneously interacts with two molecular velocity groups with opposite signs. This cancels the influence of the pump frequency drift on the position of the probe lines and improves the frequency precision by more than an order of magnitude. The high SNR and frequency precision allow using two independent methods of assigning the rotational quantum number of the final state of the probe transitions without the need to rely on theoretical predictions. The first is based on the differences in OODR probe line intensities measured with parallel and perpendicular relative pump/probe polarizations, which arise from the sample birefringence induced by the pump laser. The second is using combination differences, i.e., reaching the same final state by different combinations of pump and probe frequencies. We show that these two methods are in agreement with each other, and the assignments are confirmed by theoretical predictions from the TheoReTS/HITEMP database. This opens up the sensitive detection and unambiguous assignment of hot-band transitions of molecules for which theoretical predictions are missing or inaccurate.

## Results
### Experimental setup and procedures

To demonstrate cavity-enhanced frequency comb OODR spectroscopy we use a high-power 3.3 μm (3000 cm$^{-1}$) CW pump and a 1.68 μm (5950 cm$^{-1}$)-centered frequency comb probe, as shown in Fig. 1a. The pump frequency is Lamb-dip locked to a CH$_4$ transition from a vibrational ground state and populates a selected assigned state in the $\nu_3$ band, while the comb simultaneously probes sub-Doppler hot-band transitions from the pumped $\nu_3$ state and Doppler-broadened transitions from the ground state, as shown in Fig. 1b. The final states reached by the sub-Doppler probe transitions have term values in the 8990–9010 cm$^{-1}$ range and belong to different sub-bands within the triacontad polyad of methane, where the dominating band is $3\nu_3 \leftarrow \nu_3$. The Doppler-broadened absorption is dominated by the $2\nu_3$ cold band.

The sample of 50 mTorr of pure CH$_4$ is contained in an 80-cm-long cavity resonant with the comb probe (187 MHz free spectral range, FSR) but highly transmitting and thus non-resonant with the pump. The cavity finesse varies between 5000 and 8500 in the 1672–1692 nm (5910–5980 cm$^{-1}$) range (see Supplementary Note 1). The comb is locked to the cavity using the two-point Pound–Drever–Hall (PDH) stabilization scheme[23] (see the "Methods" section for details). The repetition rate, $f_{\text{rep}}$, is locked to a tunable source referenced to a GPS-disciplined Rb oscillator, and the carrier-envelope offset frequency, $f_{\text{ceo}}$, is monitored using an $f$–$2f$ interferometer. The pump and probe beams are combined in front of the cavity using a dichroic mirror and their relative polarization is adjusted to be either parallel or perpendicular using a half-wave plate in the pump beam. The pump passes once through the cavity and the transmitted power is monitored using a power meter after a second dichroic mirror that separates the pump and probe beams after the cavity.

The transmitted comb has a bandwidth of 1.5 THz (15 nm, 50 cm$^{-1}$) limited by cavity mirror dispersion, and its center frequency can be tuned anywhere within the bandwidth of the incident comb by a proper choice of the locking points in the two-point PDH scheme. The comb beam is led via a polarization-maintaining optical fiber to a fast-scanning Fourier transform spectrometer with auto-balanced

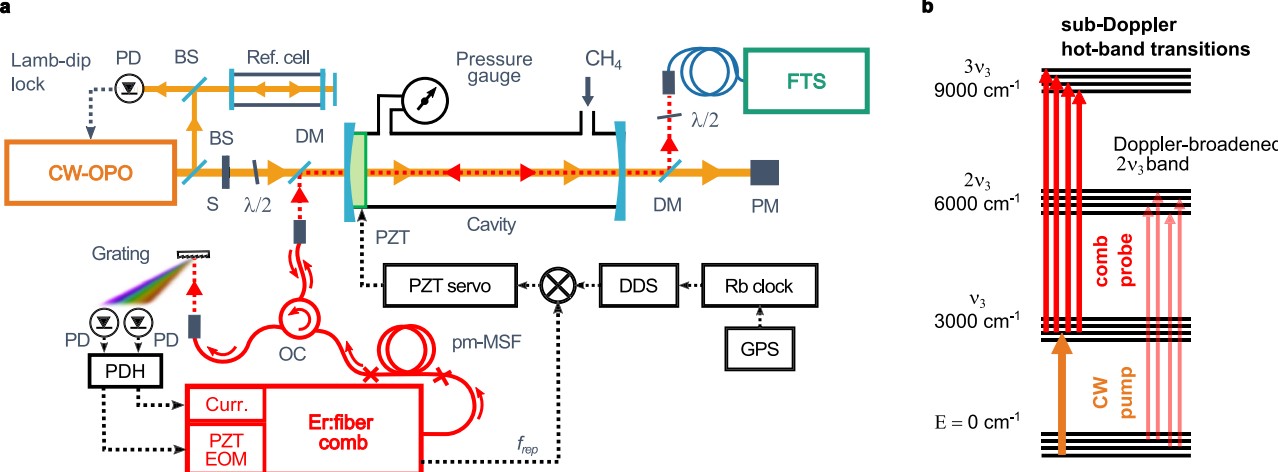

**Fig. 1 | The experimental setup and the energy levels involved in OODR spectroscopy. a** Experimental setup. CW-OPO continuous-wave optical parametric oscillator, BS beam splitter, S shutter, λ/2 half-wave plate, DM dichroic mirrors, PZT piezoelectric transducer, FTS Fourier transform spectrometer, PM power meter, pm-MSF polarization-maintaining microstructured silica fiber, OC optical circulator, PD photodiodes, Curr. current input of the comb oscillator, EOM electro-optic modulator, DDS direct digital synthesizer. **b** Simplified representation of the vibrational bands of methane addressed by the CW pump (orange) and the comb probe (red).

detection[24]. Spectra are recorded at different $f_{rep}$ values and interleaved to yield a sample point spacing of 2 MHz in the optical domain (see the "Methods" section for details). Comb-mode-limited resolution is obtained using the method of refs. [25,26], which relies on matching the nominal resolution of the spectrometer to the $f_{rep}$ (see Supplementary Note 4). To remove the background originating from the comb envelope and the Doppler-broadened absorption of the $2\nu_3$ band, at each $f_{rep}$ step we use a shutter to acquire spectra with and without the CW pump excitation and take their ratio. The slowly varying baseline in the normalized spectrum, arising from intensity fluctuations of the cavity transmission, is removed using the cepstral method[27]. The total acquisition time of one normalized and interleaved spectrum containing 750,000 sampling points spaced by 2 MHz is 16.7 min.

We recorded spectra with the pump consecutively locked to three transitions in the $\nu_3$ band starting from the same level in the ground state with rotational quantum number $J = 2$, namely the P(2, $F_2$), Q(2, $F_2$), and R(2, $F_2$) transitions. For the P(2, $F_2$) and Q(2, $F_2$) pump transitions, we recorded 5 series of $f_{rep}$ scans with both pump polarizations, while for the R(2, $F_2$) pump transition we recorded 5 series with perpendicular polarization, and 45 series with parallel polarization (see Supplementary Note 2). The pump frequencies and the corresponding comb probe coverage are summarized in Table 1. The OODR probe transitions were found in each interleaved spectrum using a peak detection routine similar to that used in ref. 22.

### Sensitivity

A narrow section of the interleaved and normalized probe spectrum recorded with the pump locked to the $\nu_3$ R(2, $F_2$) transition and averaged 5 times is shown by the black curve in Fig. 2, revealing two sub-

Doppler OODR probe transitions on a flat baseline. The noise on the baseline is on average $\sigma = 4.7 \times 10^{-3}$, which translates to the lowest detectable absorption coefficient, $\alpha_{min} = \sigma/L_{eff}$, of $1.5 \times 10^{-8}$ cm$^{-1}$ at 1.4 h, where the effective cavity length, $L_{eff}$, is given by $2FL/\pi$, where, in turn, $L$ is the cavity length, and $F$ is the cavity finesse equal to 6000 at 5929 cm$^{-1}$, while 1.4 h is the measurement time of 5 interleaved spectra, $\tau$. The noise-equivalent absorption sensitivity, defined as $\alpha_{min}\tau^{1/2}$, is $1.1 \times 10^{-6}$ cm$^{-1}$ Hz$^{-1/2}$, which is a factor of 700 better than in the previous measurement employing the liquid-nitrogen-cooled single-pass cell[21,22]. At 110 K, the temperature of the previous single-pass-cell measurement, the pump signal for $J = 2$ lines at a given pressure is 14.6 times stronger than at room temperature (see Supplementary Note 5). This implies that the room-temperature cavity allows the detection of 700/14.6 = 50 times weaker probe transitions than before at the same pressure and acquisition time.

The figure of merit, defined as $\alpha_{min}(\tau/M)^{1/2}$, where $M$ is the number of spectral elements, is $1.3 \times 10^{-9}$ cm$^{-1}$ Hz$^{-1/2}$ per spectral element, on par with what previously has been achieved in the same spectral region using cavity-enhanced dual-comb spectroscopy[28] or in the mid-infrared using cavity-enhanced comb-based dispersive spectrometer designed for physical–chemistry applications[29]. However, none of the previously demonstrated cavity-enhanced comb-based spectrometers had sub-Doppler resolution and the capability to detect hot-band transitions.

### Frequency uncertainty

The three strongest OODR probe transitions detected with the pump locked to the $\nu_3$ R(2, $F_2$) transition are shown in Fig. 3 for parallel (red markers) and perpendicular (black markers) relative pump/probe polarizations. The curves show fits of the cavity-enhanced transmission function[23] (see Supplementary Note 3), from which we retrieve the center frequencies, integrated absorptions, and widths of the probe lines. The asymmetry in the line shapes, visible in Fig. 3a and b, is caused by the offset of the comb modes from cavity resonances, which in turn is caused primarily by the dispersion of the cavity mirror coatings. This comb-cavity offset is zero close to the PDH locking points, e.g. in Fig. 3c, and increases away from them. This effect is included in the cavity transmission function and does not affect the accuracy of the center frequency determination. The residuals visible around the line centers indicate that modeling the OODR probe transitions as single Lorentzian peaks (see Supplementary Note 3) is not

### Table 1 | Pump transitions and probed ranges

| Measurement | Pump transition ($\nu_3$ band) | Pump wavenumber[31,33] [cm$^{-1}$] | Probe coverage [cm$^{-1}$] |
| --- | --- | --- | --- |
| 1 | P(2, $F_2$) | 2998.99403200(7) | 5935–5985 |
| 2 | Q(2, $F_2$) | 3018.65020715(7) | 5925–5975 |
| 3 | R(2, $F_2$) | 3048.15331810(8) | 5905–5945 |

The transitions pumped in the three measurement series, their wavenumbers from refs. 31,33. and the corresponding spectral coverage of the comb probe.

fully appropriate, and work is ongoing on improving the accuracy of the model. We note that the residuals are symmetric around the line center and thus the inaccuracy of the model does not affect the accuracy of the center frequency determination. The width of the lines is of the order of 5 MHz, dominated by power broadening caused by the pump.

To investigate the long-term stability of center frequency, we performed fits to 9 OODR probe transitions detected in the 45 consecutive spectra recorded over 12 h with the pump locked to the $v_3$ R(2, $F_2$) transition and parallel relative pump/probe polarizations. The center frequencies of the three transitions from Fig. 3, obtained from these fits, are shown in Fig. 4, offset by their mean value. The error bars show the statistical standard errors from the individual fits, which vary between 15 and 112 kHz, while the shaded area indicates one standard deviation of all values, equal to 150 kHz ($5 \times 10^{-6}$ cm$^{-1}$). We attribute the fact that the spread of the center frequencies of the fits is larger than their precision to residual uncorrected baseline drift. The 150 kHz precision of the center frequency, for lines with SNR larger than 50 (see Supplementary Note 4.2), is more than an order of magnitude better than obtained previously in the single-pass cell (1.7 MHz)[22], which confirms that the influence of the drift of the pump center frequency on the positions of the probe lines has been canceled.

We note that, based upon literature values for other CH$_4$ rovibrational transitions, the influence of the pressure and Stark shift on the probe transition frequencies is smaller than the precision of the measurement. Using the self-induced pressure shift coefficient of ($-0.017 \pm 0.003$) cm$^{-1}$/atm$^{-1}$ for methane lines in the 6000 cm$^{-1}$ range reported by Lyulin et al.[30] yields a $-33$ kHz pressure shift for the probe lines at 50 mTorr, which is below the uncertainty we report. Okubo et al.[31] reported a power shift coefficient of a sub-Doppler Lamb dip in the P(7, $E$) line of the $v_3$ band to be ($-13 \pm 17$) kHz/W for a beam radius of 0.71 mm. For the beam radius and power of pump in our experiment, this results in a ($-0.8 \pm 1.1$) kHz shift, which also is negligible.

## Line assignments

We assign the branches of the detected OODR probe transitions using two independent methods. The first method uses the fact that the ratio of the probe line intensities measured with parallel and perpendicular relative pump/probe polarizations depends on the change of rotational quantum number $J$, i.e., it is different for P, Q, and R pump and probe transitions[32]. This is because the dipole moments of both the pump and the probe transitions, for each value of the projection of the total angular momentum on the quantization axis (defined by the pump electric field), depend on the total angular momentum quantum numbers of the two states of the transition and the direction of the optical electric field. The polarization-dependent intensity ratios can thus be predicted from the transition dipoles of the pump and probe transitions[32] (see Supplementary Note 6). For example, when an R(2) transition is pumped, the parallel over perpendicular polarization integrated intensity ratios are predicted to be 1.85, 0.35, and 1.30 for a P(3), Q(3), and R(3) probe transition, respectively. For the lines shown in Fig. 3a–c, these intensity ratios are 0.4(1), 1.6(2), and 1.36(9), respectively, where the uncertainty is mainly given by the uncertainty in probe polarization (see Supplementary Note 6). This suggests the line assignment as Q(3, $F_1$), P(3, $F_1$), and R(3, $F_1$). Table 2 lists in the last column the predicted and measured intensity ratios for all combinations of pump and probe transitions detected in this work.

The second method of branch assignment is based on combination differences, i.e., cases when the same final energy state is reached by two or three different combinations of pump and probe frequencies, as is schematically shown in Fig. 5. Selection rules allow assigning the rotational quantum number $J$ of the final states depending on which probe spectra a given final state appears in. To

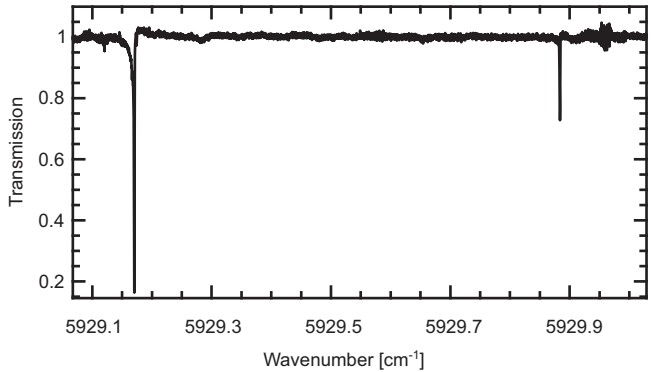

**Fig. 2 | OODR spectrum.** A narrow section of the probe spectrum measured with the pump locked to the $v_3$ R(2, $F_2$) transition and perpendicular relative pump/probe polarizations (5 averages) that contains two sub-Doppler probe transitions.

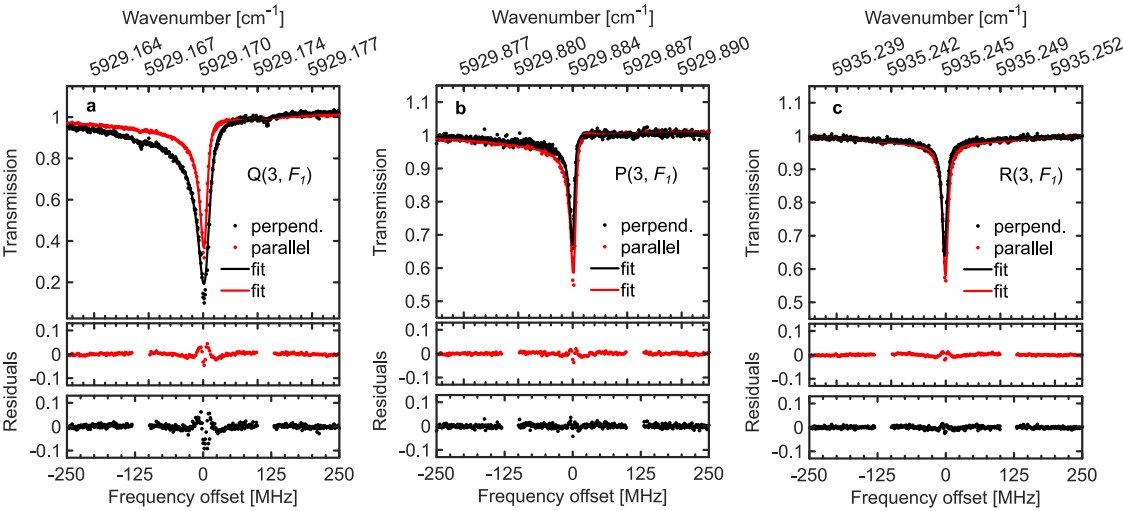

**Fig. 3 | OODR probe transitions.** Three probe transitions: **a** $3v_3 \leftarrow v_3$ Q(3, $F_1$), **b** $v_1 + 4v_2 \leftarrow v_3$ P(3, $F_1$), and **c** $v_2 + v_3 + v_4 \leftarrow v_3$ R(3, $F_1$), measured when the pump is locked to the $v_3$ R(2, $F_2$) transition. Upper windows: data taken with parallel (red markers, 5 averages) and perpendicular (black markers, 5 averages) relative pump/ probe polarizations together with fits of the cavity transmission function (solid curves). Lower windows: residuals of the fits. Line assignment—see text under Line assignments.

find the combination differences among the measured transitions, we calculate their final state term value as the sum of the ground state term value 31.4423878(8) cm⁻¹ from private communication with Hiroyuki Sasada, the pump transition frequencies from refs. 31,33 (known with kHz accuracy, see Table 1), and the measured probe transition wavenumbers (listed in Supplementary Table 2). Common final states for different combinations of pump and probe transitions are easily identified as states whose term values agree with the experimental uncertainty, while the separations between the different final states are significantly larger than the experimental uncertainty. This confirms that the experimental uncertainties are not underestimated. Table 2 lists the final state term values reached by the three probe transitions shown in Fig. 3, together with other pump/probe combinations that reach the same states—if they exist. The probe transition assignment shown in column 3 is based on the combination differences, and it is consistent with the assignment based on the intensity ratios, shown in the last column of the table. The dominant assignment of the final state in column 6 is obtained from the non-empirical effective Hamiltonian described in ref. 34.

We note that if transitions are not missed because of too low intensity, overlap with Doppler-broadened $2\nu_3$ transitions whose absorption is saturated, or because of being outside the probed spectral region, this method gives an unambiguous $J$ assignment for each final state reached.

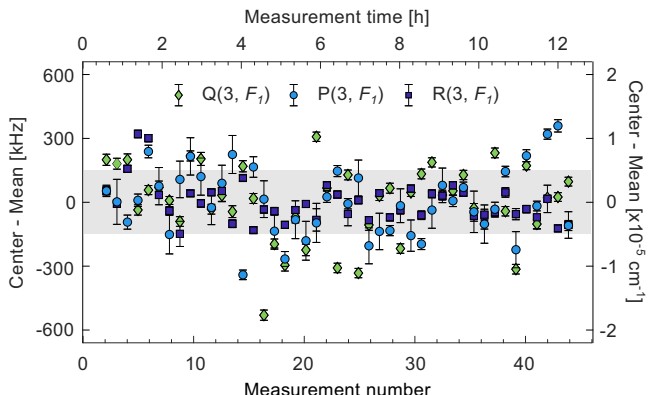

**Fig. 4 | Long-term frequency measurement.** Center frequencies (left axis) and wavenumbers (right axis) from fits to 45 consecutive measurements of the three probe transitions shown in Fig. 3, as marked in the legend, offset by their mean. The error bars show the fit precisions and the shaded area indicates one standard deviation of all values.

## Comparison to theoretical predictions

In total, we detected 21 OODR probe transitions, whose intensities span more than two orders of magnitude, as shown in Fig. 6a. 15 transitions, marked by rhombs, squares, and circles for the P(2, $F_2$), Q(2, $F_2$), and R(2, $F_2$)-pumped spectra, respectively, were measured with both relative pump/probe polarizations and could be assigned using the polarization-dependent intensity ratios, while the 6 weakest transitions, marked by stars, were observed only in the 45-times-averaged spectrum with pump locked to the $\nu_3$ R(2, $F_2$) transition and parallel pump/probe polarizations. For 9 transitions, the assignment is confirmed using combination differences.

We compare the measured line intensities and positions to predictions from the TheoReTS/HITEMP database[11]. Figure 6b shows the ratios of the experimental and predicted integrated absorptions of the probe lines, while Fig. 6c displays the differences between the center wavenumbers of the observed and the predicted transitions (see the "Methods" section for details). We note that the TheoReTS predictions are missing for two weak probe transitions, marked by the gray stars in Fig. 6a. The two outliers that are visible in the relative intensity plot, (Fig. 6b), correspond to two of the weakest detected lines, for which the predictions might be less accurate. Neglecting these two outliers, the mean intensity ratios for the lines in the P(2, $F_2$), Q(2, $F_2$), and R(2, $F_2$)-pumped spectra are constant to within 12%, 5%, and 17%, respectively, while TheoReTS states an average accuracy of 2–3% on the integrated absorption[35]. The line positions are within 1.3 cm⁻¹ from the predictions, which is roughly within the estimated TheoReTS accuracy of 1 cm⁻¹.

## Discussion

In this work, we demonstrate cavity-enhanced frequency comb OODR spectroscopy that allows the detection and assignment of sub-Doppler hot-band transitions over a wide spectral range with high absorption sensitivity and frequency precision on the 150 kHz level. Compared to the previous demonstration of comb-based OODR that employed a liquid-nitrogen-cooled single-pass cell[21,22], the use of the cavity increases by more than an order of magnitude both the absorption sensitivity (by increasing the interaction length of the probe with the sample) and frequency precision (by eliminating the frequency shift caused by the residual drift in the pump frequency) while allowing operation at room temperature. The lack of cooling improves the accuracy of intensity measurements since there is a negligible temperature gradient in the cell. The room temperature operation will allow measurements of hot-band transitions from highly rotationally excited levels that are not accessible for the pump at liquid nitrogen temperatures, and for which theoretical predictions have not yet been verified. Most importantly, the technique can now be applied to

## Table 2 | Parameters of the OODR transitions

| Pump transition ($\nu_3$ band) | Pump transition upper state term value [cm⁻¹] | Probe transition | Probe transition wavenumber [cm⁻¹] | Final state term value [cm⁻¹] | Final state assignment[34] | Polarization-dependent intensity ratio | |
|---|---|---|---|---|---|---|---|
| | | | | | | Predicted[32] | Measured |
| P(2, $F_2$) | 3030.4364198(8) | R(1, $F_1$) | 5979.042972(4) | 9009.479391(4) | $\nu_1 + 4\nu_2$ ($A_1$) | 1.01 | 0.98(6) |
| Q(2, $F_2$) | 3050.0925950(8) | Q(2, $F_1$) | 5959.386794(5) | 9009.479389(5) | $\nu_1 + 4\nu_2$ ($A_1$) | 2.00 | 1.8(3) |
| R(2, $F_2$) | 3079.5957059(8) | P(3, $F_1$) | 5929.883689(4) | 9009.479394(4) | $\nu_1 + 4\nu_2$ ($A_1$) | 1.85 | 1.6(2) |
| Q(2, $F_2$) | 3050.0925950(8) | R(2, $F_1$) | 5958.673570(6) | 9008.766165(6) | $3\nu_3$ ($F_1$) | 0.80 | 0.83(7) |
| R(2, $F_2$) | 3079.5957059(8) | Q(3, $F_1$) | 5929.170466(4) | 9008.766172(4) | $3\nu_3$ ($F_1$) | 0.35 | 0.4(1) |
| R(2, $F_2$) | 3079.5957059(8) | R(3, $F_1$) | 5935.245195(4) | 9014.840901(4) | $3\nu_2 + \nu_3 + \nu_4$ ($F_2$) | 1.30 | 1.36(9) |

Parameters of the transitions shown in Fig. 5 allow the assignment of the rotational quantum number of the final state via combination differences. Column 1: Pump transition. Column 2: Pump transition upper state term value, calculated as a sum of pump transition frequencies from refs. 31,33 and ground state term value from private communication with Hiroyuki Sasada. Column 3: Assignment of the probe transition based on combination differences. Column 4: Experimental probe transition wavenumber. Column 5: Final state term value. Column 6: Final state dominant assignment from the non-empirical effective Hamiltonian described in ref. 34. Column 7: Predicted (from ref. 32) and experimental intensity ratios for parallel and perpendicular relative pump/probe polarizations.

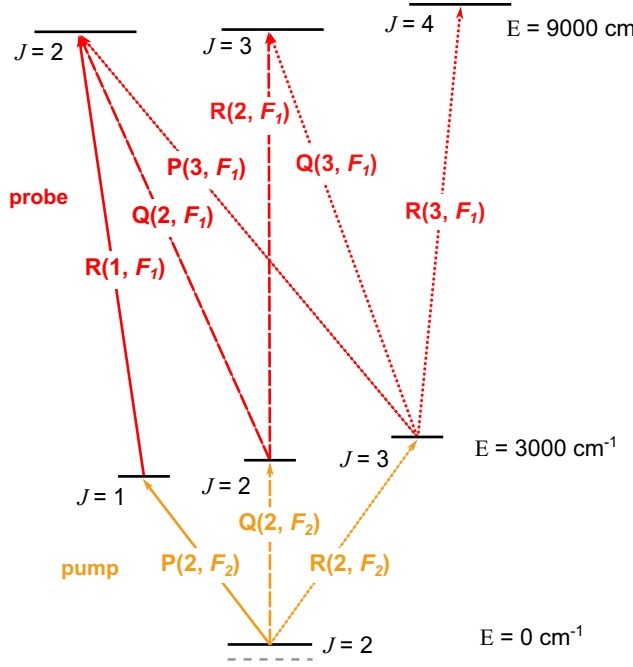

**Fig. 5 | Final state assignment using combination differences.** Simplified illustration of the observed combination differences used to assign the final states of probe transitions. The solid, dashed, and dotted lines illustrate the transitions corresponding to the case when the P(2, $F_2$), Q(2, $F_2$), and R(2, $F_2$) transitions in the $\nu_3$ band are pumped, respectively. Transitions reaching final states with $J = 0$ and 1 are out of the measured probe range in this work and are not shown.

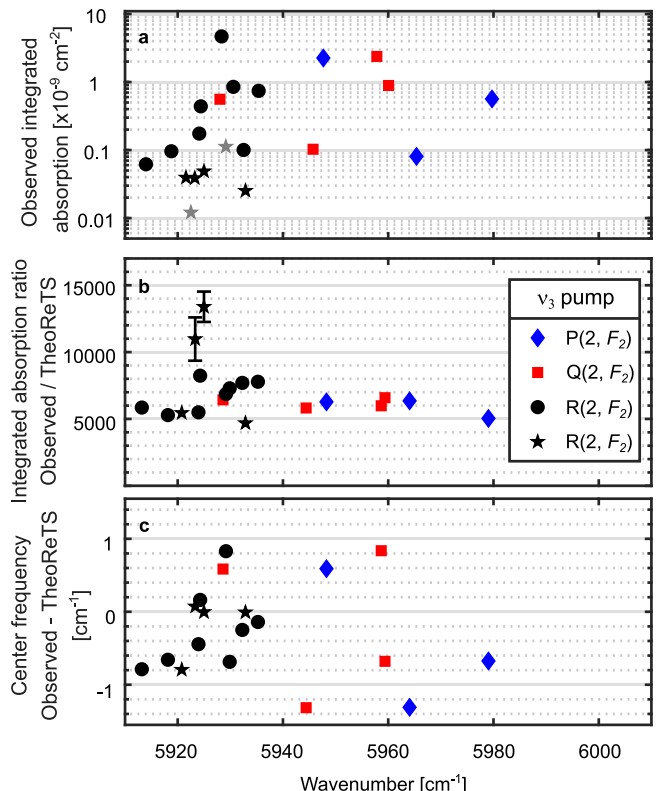

**Fig. 6 | Comparison to the TheoReTS/HITEMP database. a** Integrated absorption of all measured OODR probe transitions on a logarithmic scale. **b** Ratios of the experimental integrated absorption from **a** and the integrated absorption predicted at 296 K and 50 mTorr using line intensities from the TheoReTS/HITEMP database. The error bars for all lines but two are of the order of the marker size and therefore not plotted. **c** Center wavenumbers of the OODR probe transitions compared to predictions from the TheoReTS/HITEMP database. The experimental uncertainties are negligible on this scale. The rhombs, squares, and circles correspond to the different pumped transitions in the $\nu_3$ band, as indicated in the legend. The stars indicate the 6 weak transitions found only in the dataset averaged 45 times with pump locked to the $\nu_3$ R(2, $F_2$) transition. The two transitions marked by gray stars lack assignment in TheoReTS/HITEMP and are therefore missing in **b** and **c**.

molecules other than methane that would condense in a liquid-nitrogen-cooled cell.

The broad spectral coverage of the comb probe, the high-frequency precision, and the high SNR allow using two independent methods of assigning the rotational quantum number of the final states of the probe transitions, namely combination differences and polarization-dependent intensity ratios. The two methods are in agreement with each other, and the assignments are confirmed by predictions from the TheoReTS database. This implies that, in the future, it will be possible to assign transitions that lack theoretical predictions.

Cavity-enhanced comb-based OODR spectroscopy allows measuring and assigning individual hot-band transitions that would be indiscernible in high-temperature absorption spectra measured at local thermodynamic equilibrium. Using different combinations of CW pump and comb probe frequencies, the technique will enable systematic measurements and assignments of weak hot-band transitions of many molecules over a broad spectral range. The energy levels determined in this work are involved in hot-band transitions spanning the entire JWST observation window. Thus, an OODR measurement in one range has an impact on the accuracy of hot-band predictions across the entire infrared range covered by the JWST instruments. Reference data provided by this technique will lead to improved and new theoretical predictions of high-temperature spectra of many molecular species, needed to confirm (or disprove) their detections in future astrophysical observations.

## Methods
### Pump and probe frequency stabilization and cavity mode matching
The pump is the idler of a singly-resonant CW optical parametric oscillator (CW-OPO, Aculight, Argos 2400 SF, module C). Its frequency is stabilized to the center of the Lamb dip in the selected $CH_4$ transition in the $\nu_3$ band using a frequency-modulated error signal (modulation

frequency 60 MHz) from a reference cell, as described in detail in ref. 22. The pressure in the Lamb dip cell was 30 mTorr for locking to the Q(2, $F_2$) and R(2, $F_2$) transitions, and 190 mTorr for locking to the weaker P(2, $F_2$) transition.

The probe is an amplified Er:fiber frequency comb (Menlo Systems, FC1500-250-WG) with an $f_{rep}$ of 250 MHz. The comb spectrum is shifted to cover 6 THz around 1.68 μm using a polarization-maintaining microstructured silica fiber[36]. The comb is locked to the cavity using the two-point PDH stabilization scheme[23], where two error signals are derived from the light reflected from the cavity, picked up using a fiber optical circulator, and dispersed by a free-space reflection grating. Two selected ranges of the dispersed light, referred to as locking points, are incident on two high-bandwidth photodetectors. Correction signals are derived from the two detectors using proportional-integral controllers and sent to the current of the oscillator pump diode, which controls $f_{rep}$ and $f_{ceo}$, and to a PZT and electro-optic modulator in the oscillator cavity that control the $f_{rep}$. Absolute frequency stability is ensured by locking $f_{rep}$ to the output of a tunable direct digital synthesizer (DDS) referenced to a GPS-disciplined Rb oscillator via actuating on the sample cavity length, similar to what was done in ref. 26. The $f_{ceo}$ is indirectly stabilized via the comb-cavity lock and monitored using an $f$–$2f$ interferometer during the acquisition of the spectra.

The 80-cm-long cavity is made of two mirrors (Layertec) with 5-m radius of curvature, with maximum reflectivity and minimum dispersion at the design wavelength of 1580 nm. The mirror transmission at the pump wavelength is 60%. The transmitted comb has a mode spacing of 750 MHz, resulting from the 4:3 ratio of the incident comb $f_{rep}$ and the cavity FSR. We note that this filtering of comb modes is not necessary for the operation of the technique, but it is a result of using a cavity from a different experiment[37], where such filtering was needed. The comb beam is mode matched to the $TEM_{00}$ transverse mode of the cavity, which has a Rayleigh range of 1.4 m at 1650 nm. To maximize the spatial overlap between the pump and probe beams in the cavity, the pump beam is mode-matched to have its waist in the middle of the cavity and the same Rayleigh range, which corresponds to a pump beam waist of 1.2 mm at 3.3 μm. The pump power incident on the sample, calculated as the power transmitted through the cavity when the pump is off-resonance, divided by the mirror transmission, is 180, 165, and 150 mW for the P(2, $F_2$), Q(2, $F_2$) and R(2, $F_2$)-pumped spectra, respectively, and the fractional pump transmission on resonance (i.e., when locked to the center of the pump transition) is 64%, 70% and 61%, respectively.

### Spectral acquisition

The probe comb spectra are measured using a home-built Fourier transform spectrometer (FTS) with auto-balanced detection, previously used in refs. 21,22,26. The optical path difference is calibrated using a stabilized 633-nm HeNe laser (Sios, SL/02/1), which has a fractional frequency stability of $5 \times 10^{-9}$ over 1 h. The comb interferograms are recorded simultaneously with the reference laser interferograms by a digital oscilloscope (National Instruments, PCI-5922) with a sampling rate of 5 MS/s and a 20-bit resolution. The comb interferograms are interpolated at the zero-crossings and extrema of the corresponding HeNe interferogram.

During the acquisition, the sample and background interferograms are measured with the pump unblocked and blocked on consecutive FTS scans using the shutter (Thorlabs, SHB1T). To record the narrow sub-Doppler transitions, the sample point spacing needs to be much smaller than the $f_{rep}$. Therefore, the comb $f_{rep}$—and with it the cavity FSR—are tuned by stepping the DDS frequency, and spectra are recorded at 387 values of $f_{rep}$ differing by 2.75 Hz, which corresponds to a shift of comb modes by 2 MHz in the optical domain. For each step, the nominal resolution of the spectrometer is matched in post-processing to the $f_{rep}$, and the frequency scale is shifted by the $f_{ceo}$, which yields spectra with a comb-calibrated frequency axis[25,26].

The acquisition time of one interferogram with a nominal resolution of 750 MHz is 1.3 s, which yields a total acquisition time of 16.7 min for one normalized and interleaved spectrum, given by $1.3 \times 2 \times 387$ s, where the factor of 2 comes from the acquisition of spectra with and without the pump at each step.

### Comparison to TheoReTS/HITEMP

All probe transitions measured with the pump locked to the $v_3$ P(2, $F_2$) and $v_3$ Q(2, $F_2$) transitions, and 4 measured with the pump locked to the $v_3$ R(2, $F_2$) transition, could be unambiguously matched to the closest strongest TheoReTS line, with the assignment of the final state agreeing with the experimental result. When more than one strong TheoReTS line was within the claimed TheoReTS accuracy of 1 cm$^{-1}$ from the measured probe transition, we used the ratios of experimental and TheoReTS integrated absorptions to verify the match. Ideally, one would use the intensities of the so-called V-type transitions in the $2v_3$ band transitions to estimate the fraction of the population transferred to the upper pump level as was done in ref. 22. However, V-type transitions are not observed in the present cavity-enhanced spectra, because the absorption of the $2v_3$ band transitions is fully saturated at 50 mTorr, and there were no other transitions with lower state rotational quantum number $J = 2$ within the probed spectral range. Therefore, we made a relative comparison of the integrated absorption of the sub-Doppler probe lines to the integrated absorption of the Doppler-broadened TheoReTS lines. We calculated the latter as a product of the TheoReTS line intensity and the sample density at 50 mTorr and 296 K, which is $1.63 \times 10^{15}$ molecules cm$^{-3}$. The experimental and predicted integrated absorptions are listed in Supplementary Table 2, while their ratios are shown in Fig. 6b. For the unambiguously matched lines in the P(2, $F_2$), Q(2, $F_2$) and R(2, $F_2$)-pumped spectra, the mean ratios are 5900(700), 6200(400), and 6100(1100), respectively. The absolute values of these ratios reflect the difference in the population of the upper pump level in the OODR experiment compared to the thermal population at room temperature. We matched another 8 lines in the R(2, $F_2$)-pumped spectrum by choosing the TheoReTS line for which the ratio of the experimental and the predicted intensities was closest to the mean for the unambiguously assigned lines. The mean intensity ratio of the lines in the R(2, $F_2$)-pumped spectrum, neglecting the two outliers visible in Fig. 6b, is 6400(1200).

## Data availability

The spectra generated and analyzed in this study, as well as data for Fig. 4, Supplementary Figs. 1 and 3, have been deposited in the Zenodo database with the identifier doi:10.5281/zenodo.10245310 (ref. 38).

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

## Acknowledgements

The authors thank Hiroyuki Sasada for providing the ground state term value from his unpublished work, and Michael Rey for providing the final state assignments from the non-empirical effective Hamiltonian described in his work, A.F. acknowledges the Knut and Alice Wallenberg Foundation (KAW 2015.0159, KAW 2020.0303) and the Swedish Research Council (2020-00238); L.R. acknowledges the French National Research Agency (ANR-19-CE30-0038-01); G.S. acknowledges the Foundation for Polish Science (POIR.04.04.00-00-434D/17-00); O.A. acknowledges the Kempe Foundation (JCK 1317.1); K.K.L. acknowledges the National Science Foundation (grant: CHE-2108458).

## Author contributions

K.K.L. and A.F. conceived the idea. V.S.O. and I.S. implemented the experiment and performed the measurements. V.S.O. analyzed and visualized the data. L.R. and K.K.L. contributed theoretical predictions and analysis tools. A.F., G.S., and O.A. provided resources. A.F. supervised the project and wrote the manuscript. V.S.O, L.R., G.S., O.A, and K.K.L. revised the manuscript.

## Funding

## Competing interests

The authors declare no competing interests.
