## [Peer Review File · Nature Communications]

Sub-Doppler optical-optical double-resonance spectroscopy using a cavity-enhanced frequency comb probeREVIEWER COMMENTS

Reviewer #1 (Remarks to the Author):

The authors perform optical-optical double resonance spectroscopy of methane using a cw laser pump and frequency comb probe. Compared to previous work, here the authors incorporate an optical enhancement cavity for the probe, which results in a noise-equivalent absorption sensitivity enhancement of 700. This enabled measurements to be performed in room temperature cell instead of a cooled cell with around 50x improved SNR. The authors use polarization dependent intensity ratios as well as combination differences to assign the observe lines and compare the positions with database predictions. Overall, the results are excellent and the paper is well written. I have a few comments below:

1. The accuracy evaluation is a bit lacking details in my opinion. I believe that the transition frequency accuracy was determined by performing a series of measurements, basically at fixed conditions albeit with small drifts present, and then looking at the scatter in these measurements. Thus, this is mostly measuring the long-term stability/precision and not accuracy. Could you include a more thorough estimation of potential errors for the line center measurements? For example:
 - a. What are the expected pressure shifts? Did you account for these (or are they negligible?)?
 - b. Are there potential Stark shifts? I would guess they are negligible, but did you do measurements at different powers to check for this?
 - c. Do you observe any birefringence in the cavity? Since the polarization is controlled with a waveplate, could a small polarization error + cavity birefringence shift the fitted line center at all?
 - d. What is the difference in line centers for parallel and perpendicular polarization measurements?
2. The SNR improvement between the current measurements and previous measurements is quoted differently in the main text (50) and in the supplemental (60). Were different lines considered in the main text and supplemental, or is this a typo?
3. The transitions shown in figure 3 have a strong asymmetry. I assume this is from cavity dispersion, which according to the supplemental is incorporated in the fitting. If so, it would be good to mention this in the main text and refer to the supplemental for more details. It would also be good to mention something about the line shape residual in the main text.
4. Why was the cavity FSR chosen to be a fractional multiple of the comb freq?
5. Can you please clarify the P,Q,R transition ordering comment on lines 298-301? I thought (from Fig 1a) that the excited state vibrational level was $3\nu_3$. Is this not true for all of the transitions?

While the measured and predicted polarization intensity ratios agree quite well overall they do differ noticeably for one line in particular in Table II (i.e., Q(2, F1)). Any thoughts about why this transition might be farther off than the others?

Reviewer #2 (Remarks to the Author):

In the manuscript, the authors performed OODR spectroscopy of the hot bands of methane employing the cavity enhanced spectroscopy and the comb-based FTS technique. They provided high-precision ro-vibrational state energy of methane at $3\nu_3$ region, which is important for astronomical observations focusing on the atmospheres of hot planets. The transition intensity ratios observed with two pump-probe polarizations are in good agreement with theoretical calculations and each transition is successfully assigned. The absolute frequency of each transition was reported with an accuracy of 150 kHz that is an order better than their previous works, and much better than the theoretical calculation which was accurate to approximately 1cm^{-1} .

The paper presents meaningful results in molecular spectroscopy contributing to astronomical observations by exploiting the unique features of frequency comb spectroscopy. On the other hand, the fact that the authors have already published two papers showing the results of OODR spectroscopy of methane (ref21 and 22) will be a point of contention for the acceptance in Nature Communications. In the two previous papers, the cooling of the sample gas was necessary to achieve detectable signal intensity with single-pass configuration. In the present paper, cavity enhanced spectroscopy allowed the measurement with room temperature that improves the accuracy by more than an order of magnitude and enabled measurements of high rotational state. However, cavity enhanced comb spectroscopy is a well-established technique, and it is difficult to make a strong claim of novelty compared to the previous papers. I conclude that the level of novelty of this paper is not sufficient for publication in Nature Communications.

The paper is basically well written, but I have a few comments. Especially the comment 2 should be considered.

1. Line155

“...simultaneously probes sub-Doppler hot-band transitions in the $3\nu_3 \leftarrow \nu_3$ region...”

This part is misleading for the reader. The authors observed a transition at about 6000cm^{-1} from the ν_3 state, but not only to the $3\nu_3$ band. I recommend to mention that there are other vibrational levels and that the authors observed them here. In the line 298, authors mentioned about it as an explanation of P, Q, R order. This part is redundant, rather the assignment of the observed vibrational state should be shown. In addition, the assignment of vibrational levels should be shown in Table II.

2. Line 244 (2.3 Frequency accuracy)

The authors claimed 150 kHz accuracy for the measurement, but as shown in Fig. S3, it depends on the SNR of the line. This should be mentioned in the main text.

The accuracy of the transition frequencies was derived by standard deviation of 45 measurements, i.e. uncertainty of the mean of the repeated measurements. For the v_3 R(2, F2) pump transitions, the estimation of the uncertainty is fine. However, for the P(2, F2)- and Q(2, F2) pumped transitions, the authors did not perform 45 times repeated measurements. If the authors are providing the center frequencies from only 5 averaged measurements for the P(2, F2)- and Q(2, F2) pumped transitions, the uncertainty must be evaluated in a different way.

3. Figure S3 in supplement

The vertical axis for green plots is the standard deviation of the repeated measurements. And the one for other plots are uncertainty of the center frequency of the fit. The line 186 is a wrong description about green plot. I recommend to add the right axis label showing “ 1σ standard deviation” as the vertical axis for green plots.

4. Table II

In the column 2, if these are numbers from a reference, please add the reference.

Reviewer #3 (Remarks to the Author):

The manuscript presents an optical-optical double resonance spectroscopy of CH₄ using a frequency comb locked to a cavity as the probe. The method could be applied to assign a condensed spectrum of molecules like CH₄. The work is publishable after considering the following suggestions and comments.

(1) I wonder if the novelty of the work is sufficient for a publication in NC. The main improvement over previous studies by the same research group (refs. 21 & 22: PRL 126: 063001 (2021); PRA 103: 022810 (2021)) is using a resonance cavity with moderate finesse locked with the comb. The main benefit realized in this work is that the sensitivity has been enhanced by a factor of about 60. Note that Fourier-transform spectroscopy probed by a comb integrated with a cavity has been implemented in quite a few groups. The difference from those studies is that an additional single-pass pump laser is used here, and the DR spectrum was taken by turning on and off the pump laser.

(2) Some important technical details need to be clarified. A major concern is the obvious asymmetry profiles observed in this work. See Figs. 2, 3, and S2. The figures show that even the asymmetric profiles have been well-modeled. However, I did not see discussions on that. The authors claimed a sum of Lorentzian and Gaussian functions was used, but that could not give an asymmetric profile at all. The authors mentioned in Supp Mat, line 92, dispersion is considered in the background, but how? In particular, the spectrum of Fig.3a shows a Fano-like feature. This should be explained explicitly.

(3) The OODR linewidth observed in this work looks quite broad, as shown in Fig.3, which was not explained in the manuscript. How is the pump laser aligned to the probe laser? Could it be a consequence of the misalignment between the pump and probe?

(4) Differences between the spectra observed with parallel and perpendicular pump/probe were used heavily in this work. Therefore, the quality of the polarization of both pump and probe beams should be given. Particularly, the beam transmitted from the cavity may change the polarization.

(5) An accuracy of 150kHz was given to the results, which mainly comes from Fig.4 and discussion in Supp Mat 4.2. However, that is mainly a statistical uncertainty. The authors should discuss if there is any possible systematic uncertainty here. In particular, is there any shift due to the asymmetric line shape?

(6) The authors claimed that the V-type DR transitions are not observed here because the $2\nu_3$ band is fully saturated in the cavity-enhanced spectra. However, the saturation problem could be easily avoided by reducing the sample pressure in the cavity. Is there any other problem or just the authors did not try?

(7) Discussion on the comparison between the I-N₂-cooled cell used in the previous work by the same research group (Refs. 21 & 22) and the room-temperature cavity used in this work reads somehow misleading in Sec.3 (lines 381- 385). More accurately, the benefit is from the use of a cavity, not from "room temperature".

Dear Reviewers,

We thank you for your careful evaluation of our manuscript “Sub-Doppler optical-optical double-resonance spectroscopy using a cavity-enhanced frequency comb probe” by de Oliveira et al. We are pleased to see you appreciated the high quality of our research and we are grateful for the comments that allowed us to improve our manuscript.

Two Reviewers expressed concern about the novelty of our work in the context of previous works using cavity-enhanced comb spectroscopy and our own previous work with comb-based double-resonance spectroscopy. Reviewer #2 wrote:

‘The paper presents meaningful results in molecular spectroscopy contributing to astronomical observations by exploiting the unique features of frequency comb spectroscopy. On the other hand, the fact that the authors have already published two papers showing the results of OODR spectroscopy of methane (ref 21 and 22) will be a point of contention for the acceptance in Nature Communications. In the two previous papers, the cooling of the sample gas was necessary to achieve detectable signal intensity with single-pass configuration. In the present paper, cavity enhanced spectroscopy allowed the measurement with room temperature that improves the accuracy by more than an order of magnitude and enabled measurements of high rotational state. However, cavity enhanced comb spectroscopy is a well-established technique, and it is difficult to make a strong claim of novelty compared to the previous papers. I conclude that the level of novelty of this paper is not sufficient for publication in Nature Communications.’

Reviewer #3 wrote:

‘I wonder if the novelty of the work is sufficient for a publication in NC. The main improvement over previous studies by the same research group (refs. 21 & 22: PRL 126: 063001 (2021); PRA 103: 022810 (2021)) is using a resonance cavity with moderate finesse locked with the comb. The main benefit realized in this work is that the sensitivity has been enhanced by a factor of about 60. Note that Fourier-transform spectroscopy probed by a comb integrated with a cavity has been implemented in quite a few groups. The difference from those studies is that an additional single-pass pump laser is used here, and the DR spectrum was taken by turning on and off the pump laser.’

Let us first address these concerns.

Cavity-enhanced frequency comb spectroscopy was first demonstrated around 15-20 years ago using dispersive detection methods¹⁻⁴, and a couple of years later using Fourier transform spectroscopy, both with the dual-comb approach⁵ and with mechanical interferometers^{6,7}. Since then, most efforts of the community focused on further development of this complex technique, resulting in many proof-of-concept demonstrations of new approaches that improved the sensitivity⁸⁻¹⁴, resolution¹⁵⁻¹⁹, spectral coverage²⁰⁻²², intensity accuracy²³⁻²⁵, acquisition time^{10,26,27} and miniaturization^{14,28}. The performance of these new approaches has most often been validated by benchmarking against previous measurements taken using other techniques. Thus, these new developments did not provide new insights into molecular spectroscopy. Some methods of cavity-enhanced comb spectroscopy found applications in trace gas sensing in fields such as environmental monitoring²⁹⁻³¹, human breath analysis^{32,33}, molecular jets³⁴, and semiconductor gas impurities³⁵. Another type of applications is in precision spectroscopy, where the resolved modes of a comb are used to study line positions and shapes of individual molecular overtone transitions of molecules such as CO^{19,36} and CO₂³⁷⁻³⁹. Most important, in the context of the present work, are the few breakthroughs in molecular spectroscopy that have so far been enabled by the combination of the wide bandwidth, the comb-mode limited resolution and the high absorption sensitivity of cavity-enhanced comb spectroscopy. This method has

provided the first broadband high-resolution spectra of electronic bands of the molecular ion $\text{HfF}^{+40,41}$ as well as the rotationally resolved broadband spectra of fundamental bands of the complex cold molecules naphthalene, adamantane and hexamethylenetetramine⁴², and C_{60} fullerene⁴³. It was also used to observe the $\text{OD} + \text{CO} \rightarrow \text{DOC O}$ reaction kinetics⁴⁴ through broadband high-resolution time-resolved spectroscopy of transient free radicals⁴⁵.

Our work is the first application of cavity-enhanced comb spectroscopy for broadband and selective detection of hot-band molecular transitions, i.e., transitions starting from excited levels. Thus, by combining the technique with the double-resonance approach, we not only make the next big technological step for cavity-enhanced comb spectroscopy, we also directly use this method to provide spectroscopic information that was not available before, and cannot be obtained using any other method. We detect hot-band transitions of methane that have not been measured before, which goes well beyond a proof-of-principle demonstration and provides the first verification of theoretical predictions of these transitions. This is also the first cavity-enhanced frequency comb spectroscopy experiment to achieve sub-Doppler resolution.

Compared to our own previous work with comb-based DR spectroscopy, let us summarize the arguments already stated in the paper: the absorption sensitivity and frequency precision are increased by more than an order of magnitude, and the technique is now applicable to a wide range of molecules, since the use of the cavity removes the need for cooling of the sample (and polyatomic gas species other than methane cannot be cooled with liquid nitrogen). Thus, it is only in combination with an enhancement cavity that comb-based double resonance spectroscopy reaches its full potential.

Let us also note that there are – to the best of our knowledge – only three groups in the world that use Fourier-transform spectroscopy probed by a comb integrated with a cavity with its full potential, i.e., with resolved comb modes. These are the groups of Jun Ye at JILA, Piotr Maslowski in Toruń, Poland, and our group. We are aware of two other demonstrations that used this technique but with spectral resolution limited by the instrument, i.e., the groups in Grenoble⁶ and in Cork^{46,47}. The combination of dual-comb spectroscopy with cavities has been demonstrated by six groups^{5,12,13,15,23,25,27}, but none of these works went beyond a proof-of-concept demonstration.

We hope that this convinces the Reviewers that our work is not incremental in the context of previous results in this field, and that it describes a new tool that opens up new capabilities in molecular spectroscopy.

- Gherman, T. & Romanini, D. Mode-locked cavity-enhanced absorption spectroscopy. *Opt. Express* **10**, 1033-1042, doi:10.1364/OE.10.001033 (2002).
- Thorpe, M. J., Moll, K. D., Jones, R. J., Safdi, B. & Ye, J. Broadband cavity ringdown spectroscopy for sensitive and rapid molecular detection. *Science* **311**, 1595-1599, doi:10.1126/science.1123921 (2006).
- Gohle, C., Stein, B., Schliesser, A., Udem, T. & Hansch, T. W. Frequency comb Vernier spectroscopy for broadband, high-resolution, high-sensitivity absorption and dispersion spectra. *Phys. Rev. Lett.* **99**, 263902, doi:10.1103/PhysRevLett.99.263902 (2007).
- Thorpe, M. J. & Ye, J. Cavity-enhanced direct frequency comb spectroscopy. *Appl. Phys. B* **91**, 397-414, doi:10.1007/s00340-008-3019-1 (2008).
- Bernhardt, B. *et al.* Cavity-enhanced dual-comb spectroscopy. *Nat. Photonics* **4**, 55-57, doi:10.1038/NPHOTON.2009.217 (2010).
- Kassi, S. *et al.* Demonstration of cavity enhanced FTIR spectroscopy using a femtosecond laser absorption source. *Spectrosc. Acta A* **75**, 142-145, doi:10.1016/j.saa.2009.09.058 (2010).
- Foltynowicz, A., Ban, T., Maslowski, P., Adler, F. & Ye, J. Quantum-noise-limited optical frequency comb spectroscopy. *Phys. Rev. Lett.* **107**, 233002, doi:10.1103/PhysRevLett.107.233002 (2011).
- Grilli, R. *et al.* Cavity-enhanced multiplexed comb spectroscopy down to the photon shot noise. *Phys. Rev. A* **85**, 051804, doi:10.1103/PhysRevA.85.051804 (2012).

- Khodabakhsh, A., Abd Alrahman, C. & Foltynowicz, A. Noise-immune cavity-enhanced optical frequency comb spectroscopy. *Opt. Lett.* **39**, 5034-5038, doi:10.1364/OL.39.005034 (2014).
- Reber, M. A. R., Chen, Y. N. & Allison, T. K. Cavity-enhanced ultrafast spectroscopy: ultrafast meets ultrasensitive. *Optica* **3**, 311-317, doi:10.1364/optica.3.000311 (2016).
- Kowzan, G. *et al.* Self-referenced, accurate and sensitive optical frequency comb spectroscopy with a virtually imaged phased array spectrometer. *Opt. Lett.* **41**, 974-977, doi:10.1364/ol.41.000974 (2016).
- Hoghooghi, N. *et al.* Broadband coherent cavity-enhanced dual-comb spectroscopy. *Optica* **6**, 28-33, doi:10.1364/OPTICA.6.000028 (2019).
- Zhang, W., Chen, X., Wu, X., Li, Y. & Wei, H. Adaptive cavity-enhanced dual-comb spectroscopy. *Phot. Res.* **7**, 883-889, doi:10.1364/PRJ.7.000883 (2019).
- Ren, X. *et al.* Dual-comb optomechanical spectroscopy. *Nat. Commun.* **14**, 5037, doi:10.1038/s41467-023-40771-3 (2023).
- Okubo, S. *et al.* Near-infrared broadband dual-frequency-comb spectroscopy with a resolution beyond the Fourier limit determined by the observation time window. *Opt. Express* **23**, 33184-33193, doi:10.1364/OE.23.033184 (2015).
- Maslowski, P. *et al.* Surpassing the path-limited resolution of Fourier-transform spectrometry with frequency combs. *Phys. Rev. A* **93**, 021802(R), doi:10.1103/PhysRevA.93.021802 (2016).
- Rutkowski, L. *et al.* Sensitive and broadband measurement of dispersion in a cavity using a Fourier transform spectrometer with kHz resolution. *Opt. Express* **25**, 21711-21718, doi:10.1364/oe.25.021711 (2017).
- Rutkowski, L., Maslowski, P., Johansson, A. C., Khodabakhsh, A. & Foltynowicz, A. Optical frequency comb Fourier transform spectroscopy with sub-nominal resolution and precision beyond the Voigt profile. *J. Quant. Spectr. Rad. Transf.* **204**, 63-73, doi:10.1016/j.jqsrt.2017.09.001 (2018).
- Kowzan, G. *et al.* Broadband Optical Cavity Mode Measurements at Hz-Level Precision With a Comb-Based VIPA Spectrometer. *Sci. Rep.* **9**, 8206, doi:10.1038/s41598-019-44711-4 (2019).
- Foltynowicz, A., Maslowski, P., Fleisher, A. J., Bjork, B. J. & Ye, J. Cavity-enhanced optical frequency comb spectroscopy in the mid-infrared - application to trace detection of hydrogen peroxide. *Appl. Phys. B* **110**, 163-175, doi:10.1007/s00340-012-5024-7 (2013).
- Khodabakhsh, A. *et al.* Fourier transform and Vernier spectroscopy using an optical frequency comb at 3-5.4 μm . *Opt Lett* **41**, 2541-2544, doi:10.1364/OL.41.002541 (2016).
- Sulzer, P. *et al.* Cavity-enhanced field-resolved spectroscopy. *Nat. Photonics* **16**, 692-697, doi:10.1038/s41566-022-01057-0 (2022).
- Charczun, D. *et al.* Dual-comb cavity-mode width and shift spectroscopy. *Measurement* **188**, 110519, doi:10.1016/j.measurement.2021.110519 (2022).
- Dubroeuq, R. & Rutkowski, L. Optical frequency comb Fourier transform cavity ring-down spectroscopy. *Opt. Express* **30**, 13594-13602, doi:10.1364/OE.454775 (2022).
- Lisak, D. *et al.* Dual-comb cavity ring-down spectroscopy. *Sci. Rep.* **12**, 2377, doi:10.1038/s41598-022-05926-0 (2022).
- Rutkowski, L. & Morville, J. Broadband cavity-enhanced molecular spectra from Vernier filtering of a complete frequency comb. *Opt. Lett.* **39**, 6664-6667, doi:10.1364/OL.39.006664 (2014).
- Fleisher, A. J., Long, D. A., Reed, Z. D., Hodges, J. T. & Plusquellic, D. F. Coherent cavity-enhanced dual-comb spectroscopy. *Opt. Express* **24**, 10424-10434, doi:10.1364/OE.24.010424 (2016).
- Sterczewski, L. A. *et al.* Cavity-enhanced Vernier spectroscopy with a chip-scale mid-infrared frequency comb. *ACS Photonics* **9**, 994-1001, doi:10.1021/acsp Photonics.1c01849 (2022).
- Mejean, G., Kassi, S. & Romanini, D. Measurement of reactive atmospheric species by ultraviolet cavity-enhanced spectroscopy with a mode-locked femtosecond laser. *Opt. Lett.* **33**, 1231-1233, doi:10.1364/ol.33.001231 (2008).
- Grilli, R. *et al.* Trace measurement of BrO at the ppt level by a transportable mode-locked frequency-doubled cavity-enhanced spectrometer. *Appl. Phys. B* **107**, 205-212, doi:10.1007/s00340-011-4812-9 (2012).
- Grilli, R. *et al.* Frequency comb based spectrometer for in situ and real time measurements of IO, BrO, NO₂, and H₂CO at pptv and ppqv levels. *Env. Sci. Techn.* **46**, 10704-10710, doi:10.1021/es301785h (2012).
- Thorpe, M. J., Balslev-Clausen, D., Kirchner, M. S. & Ye, J. Cavity-enhanced optical frequency comb spectroscopy: application to human breath analysis. *Opt. Express* **16**, 2387-2397, doi:10.1364/OE.16.002387 (2008).
- Liang, Q. *et al.* Ultrasensitive multispecies spectroscopic breath analysis for real-time health monitoring and diagnostics. *Proc. Nat. Acad. Sci.* **118**, e2105063118, doi:doi:10.1073/pnas.2105063118 (2021).
- Thorpe, M. J., Adler, F., Cossel, K. C., de Miranda, M. H. G. & Ye, J. Tomography of a supersonically cooled molecular jet using cavity-enhanced direct frequency comb spectroscopy. *Chem. Phys. Lett.* **468**, 1-8, doi:10.1016/j.cplett.2008.11.064 (2009).
- Cossel, K. C. *et al.* Analysis of trace impurities in semiconductor gas via cavity-enhanced direct frequency comb spectroscopy. *Appl. Phys. B* **100**, 917-924, doi:10.1007/s00340-010-4132-5 (2010).
- Nishiyama, A., Kowzan, G., Charczun, D., Trawiński, R. S. & Maslowski, P. Optical frequency comb-based cavity-enhanced Fourier-transform spectroscopy: Application to collisional line-shape study. *Chinese J. Chem. Phys.* **33**, 23-30, doi:10.1063/1674-0068/cjcp1911192 (2020).
- Siciliani de Cumis, M. *et al.* Tracing part-per-billion line shifts with direct-frequency-comb Vernier spectroscopy. *Phys. Rev. A* **91**, doi:10.1103/PhysRevA.91.012505 (2015).
- Siciliani de Cumis, M. *et al.* Multiplexed direct-frequency-comb Vernier spectroscopy of carbon dioxide $2\nu_1 + \nu_3$ ro-vibrational combination band. *J. Chem. Phys.* **148**, 114303, doi:10.1063/1.5008461 (2018).

- Siciliani de Cumis, M., Eramo, R., Jiang, J., Fermann, M. E. & Cancio Pastor, P. Direct comb Vernier spectroscopy for fractional isotopic ratio determinations. *Sensors* **21**, 5883, doi:10.3390/s21175883 (2021).
- Sinclair, L. C., Cossel, K. C., Coffey, T., Ye, J. & Cornell, E. A. Frequency Comb Velocity-Modulation Spectroscopy. *Phys. Rev. Lett.* **107**, 093002, doi:10.1103/PhysRevLett.107.093002 (2011).
- Cossel, K. C. *et al.* Broadband velocity modulation spectroscopy of HfF⁺: Towards a measurement of the electron electric dipole moment. *Chem. Phys. Lett.* **546**, 1-11, doi:10.1016/j.cplett.2012.06.037 (2012).
- Spaun, B. *et al.* Continuous probing of cold complex molecules with infrared frequency comb spectroscopy. *Nature* **533**, 517-520, doi:10.1038/nature17440 (2016).
- Changala, P. B., Weichman, M. L., Lee, K. F., Fermann, M. E. & Ye, J. Rovibrational quantum state resolution of the C-60 fullerene. *Science* **363**, 49-54, doi:10.1126/science.aav2616 (2019).
- Bjork, B. J. *et al.* Direct frequency comb measurement of OD + CO → DOCO kinetics. *Science* **354**, 444-448, doi:10.1126/science.aag1862 (2016).
- Fleisher, A. J. *et al.* Mid-infrared time-resolved frequency comb spectroscopy of transient free radicals. *J. Phys. Chem. Lett.* **5**, 2241-2246, doi:10.1021/jz5008559 (2014).
- Chandran, S., Mahon, S., Ruth, A. A., Braddell, J. & Gutierrez, M. D. Cavity-enhanced absorption detection of H₂S in the near-infrared using a gain-switched frequency comb laser. *Appl. Phys. B* **124**, 63, doi:10.1007/s00340-018-6931-z (2018).
- Chandran, S. *et al.* Off-axis cavity-enhanced absorption spectroscopy of ¹⁴NH₃ in air using a gain-switched frequency comb at 1.514 μm. *Sensors* **19**, 5217, doi:10.3390/s19235217 (2019).

Below, we introduce a point-by-point response (in *italics*) to all comments of the Reviewers. The changes made to the manuscript are typed in **red**. We also append the manuscript and supplementary information with all modifications highlighted.

#####

Reviewer #1:

The authors perform optical-optical double resonance spectroscopy of methane using a cw laser pump and frequency comb probe. Compared to previous work, here the authors incorporate an optical enhancement cavity for the probe, which results in a noise-equivalent absorption sensitivity enhancement of 700. This enabled measurements to be performed in room temperature cell instead of a cooled cell with around 50x improved SNR. The authors use polarization dependent intensity ratios as well as combination differences to assign the observe lines and compare the positions with database predictions. Overall, the results are excellent and the paper is well written. I have a few comments below:

1. The accuracy evaluation is a bit lacking details in my opinion. I believe that the transition frequency accuracy was determined by performing a series of measurements, basically at fixed conditions albeit with small drifts present, and then looking at the scatter in these measurements. Thus, this is mostly measuring the long-term stability/precision and not accuracy. Could you include a more thorough estimation of potential errors for the line center measurements? For example:
 - a. What are the expected pressure shifts? Did you account for these (or are they negligible?)?
 - b. Are there potential stark shifts? I would guess they are negligible, but did you do measurements at different powers to check for this?
 - c. Do you observe any birefringence in the cavity? Since the polarization is controlled with a waveplate, could a small polarization error + cavity birefringence shift the fitted line center at all?
 - d. What is the difference in line centers for parallel and perpendicular polarization measurements?

ANSWER: We thank the Reviewer for this comment, and we agree that the influence of the pressure and Stark shifts should be discussed. We have not characterized these effects in our setup, but based on values available in the literature, these effects are negligible at the current uncertainty level:

a) Lyulin et al., *JQSRT* **112**, 531, doi:10.1016/j.jqsrt.2010.10.010 (2011) reported a self-induced pressure shift coefficient for methane lines in the 5556 – 6166 cm^{-1} range to be $(-0.017 \pm 0.003) \text{ cm}^{-1}/\text{atm}^{-1}$. This yields a -33 kHz pressure shift for the OODR probe lines at 50 mTorr, the pressure of our measurement, which is below the uncertainty we report.

b) Okubo et al., *Opt. Express* **19**, 23878, doi:10.1364/oe.19.023878 (2011) reported a power shift coefficient of the Lamb dip in the P(7,E) line of the ν_3 band to be $(-0.0039 \pm 0.0049) \text{ kHz}/\mu\text{W}$. In private communication, Prof. Sasada let us know that ‘the power level is the input to the enhanced-cavity absorption cell. At the beam waist of the cavity, the optical electric field amplitude is enhanced by 17 times, and the cavity mode radius is 0.71 mm.’ Thus, this coefficient should be divided by $17^2 = 289$ to account for the cavity enhancement of the intensity, which yields $(-13 \pm 17) \text{ kHz}/\text{W}$. Since the shift is proportional to intensity, not the power, we also need to account for the difference in the cavity mode radius between their and our experiment by multiplying with a factor of $(0.71/1.2)^2 = 0.35$. Considering the power incident on the sample in our experiment, which is up to 180 mW, the Stark shift of the pump transition is $(-13 \pm 17) \text{ kHz}/\text{W} * 0.35 * 0.18\text{W} = (-0.8 \pm 1.1) \text{ kHz}$, which is negligible.

We added the following paragraph starting at line 333:

‘We note that, based upon literature values for other CH_4 ro-vibrational transitions, the influence of the pressure and Stark shift on the probe transition frequencies is smaller than the precision of the measurement. Using the self-induced pressure shift coefficient of $(-0.017 \pm 0.003) \text{ cm}^{-1}/\text{atm}^{-1}$ for methane lines in the 6000 cm^{-1} range reported by Lyulin et al.³⁰, yields a -33 kHz pressure shift for the probe lines at 50 mTorr, which is below the uncertainty we report. Okubo et al.³¹ reported a power shift coefficient of a sub-Doppler Lamb dip in the P(7,E) line of the ν_3 band to be $(-13 \pm 17) \text{ kHz}/\text{W}$ for a beam radius of 0.71 mm. For the beam radius and power of pump in our experiment, this results in a $(-0.8 \pm 1.1) \text{ kHz}$ shift, which also is negligible.’

c) and d) We have not characterized the birefringence of the cavity used in our experiment and it is unfortunately not possible anymore, as the setup has been upgraded with a new cavity since we took the measurements described in this manuscript. However, we are not aware of any process that would shift the center frequency with pump polarization. The figure below shows the difference between the center wavenumbers measured with parallel and perpendicular pump/probe polarizations for all lines, where the error bar is the 3σ weighted uncertainty reported for the transition center. The differences are randomly spread around zero and agree with zero within the 3σ uncertainty. This proves that there is no detectable systematic effect caused by birefringence, and the uncertainties are not underestimated. We made no changes to the manuscript in response to this comment.

The uncertainty in probe polarization has an influence on the determination of the polarization-dependent intensity ratios, as described under point 6 below and in response to comment 4 of Reviewer #3.

We changed all references to ‘frequency accuracy’ to either ‘frequency precision’ or ‘frequency uncertainty’.

The difference between the center wavenumbers measured with parallel and perpendicular pump/probe polarizations for all lines, where the error bar is the 3σ weighted uncertainty reported for the transition center.

- The SNR improvement between the current measurements and previous measurements is quoted differently in the main text (50) and in the supplemental (60). Were different lines considered in the main text and supplemental, or is this a typo?

ANSWER: In the main paper we quote the SNR improvement at the same pressure but different temperature, while in the supplementary we quote the SNR improvement between the two measurements, where the pressures were different (30 mTorr in the previous measurement, compared to 50 mTorr in the current work). We kept the main text unchanged, and we made the following changes in the supplementary information:

'Considering also the difference in sample pressure (50 mTorr vs 30 mTorr), the absorption coefficient of the pump transitions was 11.7 stronger in the cell than in the cavity. This implies that the improvement in the SNR can be estimated to $700/11.7 = 60$, which agrees with what is observed in the data. We note that in the main paper we state the improvement of $700/14.6 = 50$, which is valid under same pressure conditions.'

- The transitions shown in figure 3 have a strong asymmetry. I assume this is from cavity dispersion, which according to the supplemental is incorporated in the fitting. If so, it would be good to mention this in the main text and refer to the supplemental for more details. It would also be good to mention something about the line shape residual in the main text.

ANSWER: We expanded the discussion about the asymmetry in the line shape and the origin of the residuals in the main text and in the supplementary. In the main text, we added the following text starting at line 286:

'The asymmetry in the line shapes, visible in Fig. 3(a) and (b), is caused by the offset of the comb modes from cavity resonances, which in turn is caused primarily by dispersion of the cavity mirror coatings. This comb-cavity offset is zero close to the PDH locking points, e.g. in Fig. 3(c), and increases away from them. This effect is included in the cavity transmission function and does not affect the accuracy of the center frequency determination. The residuals visible around the line centers indicate that modeling the OODR probe transitions as single Lorentzian peaks (see Supplementary information 3) is not fully appropriate, and work

is ongoing on improving the accuracy of the model. We note that the residuals are symmetric around the line center and thus the inaccuracy of the model does not affect the accuracy of the center frequency determination. The width of the lines is of the order of 5 MHz, dominated by power broadening caused by the pump.'

In the Supplementary information we added the following text in Section 3:

'The fit parameters were the wavelength-dependent comb-cavity phase offset. Dispersion in the cavity mirror coatings causes a non-zero comb-cavity resonance offset away from the PDH locking points. The cavity-transmission function also contains molecular dispersion, which results in an additional shift of the cavity resonances with respect to the comb lines. This shift has a dispersive shape as a comb mode is moved across an absorption line.¹ Since this molecular contribution to the shift is small compared to the width of the cavity modes, the change in cavity transmission is approximately linear with this molecular induced shift. This effect is well understood and included in the model, and it does not shift the fitted center frequency.'

We hope that this clarifies the origin of the asymmetry and that it does not influence the accuracy of the center frequency retrieval.

4. Why was the cavity FSR chosen to be a fractional multiple of the comb f_{rep} ?

ANSWER: The 4:3 ratio between the FSR and f_{rep} is not necessary for the OODR technique. In this first demonstration, we used a cavity that was previously used in another experiment (Ref. 38), where this ratio was required. We added a note in the method section starting at line 583:

'We note that this mode filtering is not necessary for the operation of the technique, but it is a result of using a cavity from a different experiment³⁸, where such filtering was needed.'

We have already replaced this cavity with another one that has broadband mirror coatings and $\text{FSR} = f_{\text{rep}}$.

5. Can you please clarify the P,Q,R transition ordering comment on lines 298-301? I thought (from Fig 1a) that the excited state vibrational level was $3\nu_3$. Is this not true for all of the transitions?

ANSWER: We agree that the schematic figure 1(a) (now 1b) might have been misleading. However, please note that in the abstract we wrote that 'we detect and assign sub-Doppler transitions in the $3\nu_3 \leftarrow \nu_3$ spectral range of methane', and in Section 2.1 we wrote that 'the comb ... probes sub-Doppler hot-band transitions in the $3\nu_3 \leftarrow \nu_3$ region'. This is because the dominant absorption strength in the region we probed is due to the $3\nu_3$ states (which consist of 4 distinct vibrational states that split into 10 states for each rotational wavefunction when Coriolis splittings are included), but these states are mixed with other eigenstates due to anharmonic and Coriolis couplings, particularly the Fermi resonance with the bending modes, as well as Darling-Dennison coupling between the ν_3 with ν_1 modes.

In response to this comment and comment 1 of Reviewer #2 we added the dominant assignment of the detected final states in the caption of Fig. 3, in Table II in the main text, and in Table S2 in the Supplementary information. These assignments were obtained through private communication with Michael Rey, who provided them based on the Hamiltonian from

his recent work, M. Rey, *J. Chem. Phys.* **156**, 224103, doi:10.1063/5.0089097 (2022). We added this reference and the following explanations on line 414 in the main text:

‘The dominant assignment of the final state in column 6 is obtained from the non-empirical effective Hamiltonian described in Ref. ³³.’

in the caption of Table II:

‘Column 6: Final state dominant assignment from the non-empirical effective Hamiltonian described in Ref. ³³.’

and in the description of Table S2 in section 7 of Supplementary information:

‘Column 12 shows the final state dominant assignment obtained from the Hamiltonian described in Ref. ⁸.’

We also clarified the two references to the ‘ $3\nu_3 \leftarrow \nu_3$ spectral range/region’, and we changed the caption of Fig. 1(a) (now 1b). In the abstract we write on line 28:

*‘We detect and assign sub-Doppler transitions in the **spectral range of the $3\nu_3 \leftarrow \nu_3$ resonance** of methane...’*

In Section ‘Experimental setup and procedures’ we write on line 166:

*‘The pump frequency is Lamb-dip locked to a CH_4 transition from a vibrational ground state and populates a selected assigned state in the ν_3 band, while the comb simultaneously probes sub-Doppler hot-band transitions **from the pumped ν_3 state** and Doppler-broadened transitions **from the ground state**, as shown in Fig. 1(b). The final states reached by the sub-Doppler probe transitions have term values in the $8990 - 9010 \text{ cm}^{-1}$ range and belong to different sub-bands within the triacontad polyad of methane, where the dominating band is $3\nu_3 \leftarrow \nu_3$. The Doppler-broadened absorption is dominated by the $2\nu_3$ cold band.’*

In the caption of Fig. 1(b) we write:

*‘b Simplified representation of the **vibrational bands** of methane **addressed by** the CW pump (orange) and the comb probe (red).’*

We hope that these modifications clarify what final states were reached in our measurements.

6. While the measured and predicted polarization intensity ratios agree quite well overall they do differ noticeably for one line in particular in Table II (i.e., Q(2, F1)). Any thoughts about why this transition might be farther off than the others?

ANSWER: We thank the Reviewer for this comment, which – together with comment 1 of Reviewer #3 – prompted us to investigate more closely the influence of the polarization uncertainty on the measured intensities and their ratios. In Supplementary information 6, now called ‘Predicted and experimental polarization-dependent intensity ratios’, we added an entire section that explains how the ellipticity of the probe polarization translates to the uncertainty of line intensities and their ratios. This section reads:

‘The accuracy of the measured polarization-dependent intensity ratios was limited by the quality of the probe polarization. This polarization was slightly elliptical because of the propagation in the nonlinear fiber and the fiberized optical circulator. 4.2% of the probe

power in front of the cavity was along the minor axis, which corresponds to an ellipticity of 0.2. Compared to that, the pump polarization was purely linear. To calculate the uncertainties in the measured intensities and their ratios caused by the ellipticity of the probe polarization, we assumed that the polarizations of the pump and probe are linear but offset by an angle $\varphi = 0.2$ from being perfectly parallel or perpendicular.

For the integrated intensities measured with parallel and perpendicular relative pump/probe polarizations, I_{\parallel} and I_{\perp} , we assumed a relative error of $\sin^2(\varphi) = 4.2\%$ caused by the ellipticity of the probe light.

To evaluate the uncertainty in the measured polarization-dependent intensity ratios, we compared the intensity ratios predicted for perfectly parallel and perpendicular pump and probe polarizations, given by

$$R_{\parallel/\perp}(0) = \frac{I_{\parallel}}{I_{\perp}}, \quad (1)$$

with ratios predicted for an angle φ , given by

$$R_{\parallel/\perp}(\varphi) = \frac{I_{\perp} \sin^2(\varphi) + I_{\parallel} \cos^2(\varphi)}{I_{\perp} \cos^2(\varphi) + I_{\parallel} \sin^2(\varphi)} = \frac{R_{\parallel/\perp}(0) + \sin^2(\varphi)[1 - R_{\parallel/\perp}(0)]}{1 - \sin^2(\varphi)[1 - R_{\parallel/\perp}(0)]}. \quad (2)$$

We then took twice the difference $2|R_{\parallel/\perp}(\varphi) - R_{\parallel/\perp}(0)|$, given by

$$2|R_{\parallel/\perp}(\varphi) - R_{\parallel/\perp}(0)| = \frac{2\sin^2(\varphi)[1 - R_{\parallel/\perp}^2(0)]}{1 - \sin^2(\varphi)[1 - R_{\parallel/\perp}(0)]}, \quad (3)$$

as the uncertainty in the measured polarization-dependent intensity ratio introduced by the probe ellipticity φ . We note that this uncertainty is smallest for ratios closest to 1, and larger for ratios that are farther from 1.

We added the uncertainty described above to the intensities and ratios reported in text (line 368), in Table II and Table S2. In Supplementary information 7 we added the following information:

'The uncertainty [in intensities] is a combination of the fit uncertainty, the uncertainty in the finesse determination (3%), and polarization of the probe (4.2%). Column 6 provides the ratio of the line intensities measured with the parallel and perpendicular relative pump/probe polarization. The uncertainty is a combination of the fit uncertainty of the two integrated absorptions, and the uncertainty caused by the ellipticity of the probe given by Eq. (3) above. The uncertainty in the finesse is not taken into account, since the contribution from the finesse cancels when taking the ratio of the two intensities.'

We note that the work reporting the predicted intensity ratios has been accepted for publication in *J. Chem. Phys.* and is available on arXiv, K.K. Lehmann, Polarization-dependent intensity ratios in double resonance spectroscopy, arXiv:2308.09828, doi:10.48550/arXiv.2308.09828 (2023), and we inserted reference to it in main text (ref. 32) and in the Supplementary information (ref. 6). We noticed a typo in one of the predicted

intensity ratios, for the case of R pump and P probe, and we corrected it (it was 1.57, but it should be 1.85).

We also note that we recalculated the intensities of the 6 weakest lines that were measured only with parallel pump/probe polarization to the weighted mean using the predicted polarization-dependent intensity ratios. This is explained in Supplementary information 7:

‘For the 6 weakest lines that were measured only with parallel pump/probe polarization, we recalculated the measured absorption to the weighted mean using the predicted polarization-dependent intensity ratios.’

We updated those intensities in Table S2 and in Fig. 6(a) and (b). In particular, this makes the two outliers in Fig. 6(b) closer to the mean. We also updated the mean ratios on lines 569 and 577.

#####

Reviewer #2:

In the manuscript, the authors performed OODR spectroscopy of the hot bands of methane employing the cavity enhanced spectroscopy and the comb-based FTS technique. They provided high-precision ro-vibrational state energy of methane at $3\nu_3$ region, which is important for astronomical observations focusing on the atmospheres of hot planets. The transition intensity ratios observed with two pump-probe polarizations are in good agreement with theoretical calculations and each transition is successfully assigned. The absolute frequency of each transition was reported with an accuracy of 150 kHz that is an order better than their previous works, and much better than the theoretical calculation which was accurate to approximately 1 cm^{-1} .

The paper presents meaningful results in molecular spectroscopy contributing to astronomical observations by exploiting the unique features of frequency comb spectroscopy. On the other hand, the fact that the authors have already published two papers showing the results of OODR spectroscopy of methane (ref 21 and 22) will be a point of contention for the acceptance in Nature Communications. In the two previous papers, the cooling of the sample gas was necessary to achieve detectable signal intensity with single-pass configuration. In the present paper, cavity enhanced spectroscopy allowed the measurement with room temperature that improves the accuracy by more than an order of magnitude and enabled measurements of high rotational state. However, cavity enhanced comb spectroscopy is a well-established technique, and it is difficult to make a strong claim of novelty compared to the previous papers. I conclude that the level of novelty of this paper is not sufficient for publication in Nature Communications.

We thank the Reviewer for expressing these concerns. We addressed them on the first two pages of this response letter.

The paper is basically well written, but I have a few comments. Especially the comment 2 should be considered.

1. Line155

“...simultaneously probes sub-Doppler hot-band transitions in the $3\nu_3 \leftarrow \nu_3$ region...”

This part is misleading for the reader. The authors observed a transition at about 6000 cm^{-1} from the ν_3 state, but not only to the $3\nu_3$ band. I recommend to mention that there are other vibrational levels and that the authors observed them here. In the line 298, authors mentioned about it as an explanation of P, Q, R order. This part is redundant, rather the

assignment of the observed vibrational state should be shown. In addition, the assignment of vibrational levels should be shown in Table II.

ANSWER: We followed the Reviewer's recommendation, which is also in line with comment 5 of Reviewer #1. We clarified that the $3\nu_3$ band is the dominating one in the measured region, and we added the assignments of the final states to Fig. 3 and Tables II and S2. These changes are detailed in response to comment 5 of Reviewer #1 above. In addition, we removed the text 'We note that these probe transitions reach states in different vibrational bands in the 9000 cm^{-1} range, which is why the P, Q and R lines do not appear in increasing wavenumber order.'

2. Line 244 (2.3 Frequency accuracy)

The authors claimed 150 kHz accuracy for the measurement, but as shown in Fig. S3, it depends on the SNR of the line. This should be mentioned in the main text.

The accuracy of the transition frequencies was derived by standard deviation of 45 measurements, i.e. uncertainty of the mean of the repeated measurements. For the ν_3 R(2, F₂) pump transitions, the estimation of the uncertainty is fine. However, for the P(2, F₂)- and Q(2, F₂) pumped transitions, the authors did not performed 45 times repeated measurements. If the authors are providing the center frequencies from only 5 averaged measurements for the P(2, F₂)- and Q(2, F₂) pumped transitions, the uncertainty must be evaluated in a different way.

ANSWER: We added a note on line 327 in the main text that clarifies that the 150 kHz precision is valid for lines with SNR>50:

*'The 150 kHz **precision** of the center frequency, for lines with SNR larger than 50 (see **Supplementary information 4.2**), is more than an order of magnitude better than...'*

*The long-term measurement with pump locked to the ν_3 R(2, F₂) transition revealed that the standard deviation of the center frequency retrieved from consecutive spectra is systematically higher than the fit precision (as shown by green squares and black diamonds in Fig. S3) and that it clearly depends on the signal-to-noise of the lines. This is caused by fluctuations in the baseline that depend on the stability of the comb envelope and the parameters of the baseline removal procedure, but do not depend on the pump transition. Since the former (the comb envelope and parameters of the baseline removal procedure) were similar in all measurements, we can apply the model developed from one spectrum to the other spectra. As a metric, we used the SNR of the line from a single measurement. We believe this might have been unclear in the previous version of the text in the Supplementary information 4.2, now called 'Frequency **uncertainty** model'. We made the following changes to the text, that we hope clarify the process:*

*'Figure S3 shows the 1σ standard deviation of the center frequencies from the long-term measurement (green markers, **left axis**) and the fit precision of all probe lines detected in the five combined P(2, F₂), Q(2, F₂), and R(2, F₂)-pumped spectra (blue, red, and black markers, respectively), as a function of the SNR of the line **in a single measurement**. The SNR was calculated as the ratio of the peak absorption (1-transmission) and the standard deviation of the residuals (excluding the FM sidebands and the line center due to the mismatch between the model and the measurement observed for high-SNR lines). For the long-term series (**green markers**), the SNR was taken as the mean SNR of the 45 **consecutive** spectra, where the horizontal error bar is the standard deviation of the mean. For **the probe lines fitted in the***

five combined spectra (blue, red, and black markers, respectively, right axis), the standard deviation of the residuals corresponds to the noise in a single measurement rather than an average of 5 measurements, since the data are interleaved rather than averaged (as described in Section 2).

Since the uncertainty from the long-term measurement is systematically larger than the fit precision for all lines, we developed a model for the frequency uncertainty based on the SNR dependence of the standard deviation of the long-term measurements. The blue curve in Fig. S3 shows a fit of a model function $U = (a^2 + b^2 / \text{SNR}^2)^{1/2}$ to the observed relation between the uncertainty and the SNR from the 45-measurement series, where a and b are fitting parameters. We used this function to estimate the center frequency uncertainty for all detected lines based on their single-measurement SNR evaluated as described above. Since the baseline fluctuations, to which we attribute the increased frequency uncertainty, do not depend on the choice of the pump transition, we applied the model also to the lines that were not measured 45 times, i.e., the P(2, F₂)- and Q(2, F₂)-pumped lines, and the 6 weakest R(2, F₂)-pumped lines visible only in the spectrum averaged 45 times.'

3. Figure S3 in supplement

The vertical axis for green plots is the standard deviation of the repeated measurements. And the one for other plots are uncertainty of the center frequency of the fit. The line 186 is a wrong description about green plot. I recommend to add the right axis label showing "1 σ standard deviation" as the vertical axis for green plots.

ANSWER: We thank the Reviewer for spotting this inconsistency. We added a second vertical axis to this plot, and the green markers are now shown on the left axis as '1 σ standard deviation', and the other markers are shown on the right axis as 'fit uncertainty'.

4. Table II

In the column 2, if these are numbers from a reference, please add the reference.

ANSWER: In the main text, starting at line 380, we described that 'we calculate the final state term value as the sum of the ground state term value 31.4423878(8) cm⁻¹ from Ref. ³⁴, the pump transition frequencies from Refs. ^{31,35} (known with kHz accuracy, see Table I), and the measured probe transition wavenumbers (listed in Supplementary Table S2 in the Supplementary information 7)'. In Table II, we summed the first two contributions, i.e., the ground state term value from Ref. 34 and the pump transition frequencies from Refs. 31 and 35, and presented them as the pump upper state term value. We added these references in the caption of Table II:

'Column 2: Pump transition upper state term value, calculated as a sum of pump transition frequencies from Refs ^{31,35} and ground state term value from Ref. ³⁴.'

We also added 'transition' in the header of column 2:

'Pump transition upper state term value [cm⁻¹]'

#####

Reviewer #3:

The manuscript presents an optical-optical double resonance spectroscopy of CH₄ using a frequency comb locked to a cavity as the probe. The method could be applied to assign a condensed spectrum of molecules like CH₄. The work is publishable after considering the following suggestions and comments.

1. I wonder if the novelty of the work is sufficient for a publication in NC. The main improvement over previous studies by the same research group (refs. 21 & 22: PRL 126: 063001 (2021); PRA 103: 022810 (2021)) is using a resonance cavity with moderate finesse locked with the comb. The main benefit realized in this work is that the sensitivity has been enhanced by a factor of about 60. Note that Fourier-transform spectroscopy probed by a comb integrated with a cavity has been implemented in quite a few groups. The difference from those studies is that an additional single-pass pump laser is used here, and the DR spectrum was taken by turning on and off the pump laser.

ANSWER: We thank the Reviewer for expressing these concerns. We addressed them on the first two pages of this response letter.

2. Some important technical details need to be clarified. A major concern is the obvious asymmetry profiles observed in this work. See Figs. 2, 3, and S2. The figures show that even the asymmetric profiles have been well-modeled. However, I did not see discussions on that. The authors claimed a sum of Lorentzian and Gaussian functions was used, but that could not give an asymmetric profile at all. The authors mentioned in Supp Mat, line 92, dispersion is considered in the background, but how? In particular, the spectrum of Fig.3a shows a Fano-like feature. This should be explained explicitly.

ANSWER: In response to this comment, and comment 3 of Reviewer #1, we explain now in the main text and the supplementary that this asymmetry is caused by the well-understood effects of cavity mirror and molecular dispersion. Please see the answer to comment 3 of Reviewer #1 for the list of changes made to the manuscript.

3. The OODR linewidth observed in this work looks quite broad, as shown in Fig. 3, which was not explained in the manuscript. How is the pump laser aligned to the probe laser? Could it be a consequence of the misalignment between the pump and probe?

ANSWER: The main contribution to the line broadening was explained in Supplementary information 6. We moved this information to the main text, line 299:

'The width of the lines is of the order of 5 MHz, dominated by power broadening caused by the pump.'

We used a camera to optimize the overlap of the pump and probe beams in front of and behind the cavity. This ensures that any misalignment is a small fraction of the pump beam diameter, i.e. well below 1 mm, which should then be divided by the distance between the two camera positions, which was more than 1 m. Thus, the broadening the misalignment could cause is of the order of 10^{-4} (the misalignment angle in radians) of the Doppler width, which is negligible with respect to the power broadening.

4. Differences between the spectra observed with parallel and perpendicular pump/probe were used heavily in this work. Therefore, the quality of the polarization of both pump and probe beams should be given. Particularly, the beam transmitted from the cavity may change the polarization.

ANSWER: We thank the Reviewer for this comment, which – together with comment 6 of Reviewer #1 – made us look closer into the influence of pump and probe polarization on the measured intensities and their polarization-dependent ratios. We now discuss this issue in detail in Supplementary information 6, and we reevaluated the uncertainties reported for intensities in Tables II and S2. Please see the detailed description of all changes made in response to comment 6 of Reviewer #1 above.

5. An accuracy of 150 kHz was given to the results, which mainly comes from Fig. 4 and discussion in Supp Mat 4.2. However, that is mainly a statistical uncertainty. The authors should discuss if there is any possible systematic uncertainty here. In particular, is there any shift due to the asymmetric line shape?

ANSWER: We agree that the uncertainty we provide for the line positions is the precision, not the accuracy. We changed all references to ‘frequency accuracy’ to either ‘frequency precision’ or ‘frequency uncertainty’. Please see the answer to comment 1 of Reviewer #1 for a discussion of the negligible influence of the Stark and pressure shifts. The asymmetric line shape does not cause a shift of the center frequency, as explained in the answer to comment 3 of Reviewer #1. Changes made to the manuscript are described in response to comments 1 and 3 of Reviewer #1.

6. The authors claimed that the V-type DR transitions are not observed here because the $2\nu_3$ band is fully saturated in the cavity-enhanced spectra. However, the saturation problem could be easily avoided by reducing the sample pressure in the cavity. Is there any other problem or just the authors did not try?

ANSWER: It is true that at a lower pressure the absorption of the $2\nu_3$ band transitions would not be saturated, but that would also decrease the signal of the probe transitions. In fact, the OODR signal is maximized at a pressure at which 80% of the pump power is absorbed, as we determined in our previous work, Phys. Rev. Lett. 126, 063001 (2021). In this experiment, we already operated at a pressure lower than optimum, with pump absorption of only 30-40%. This is because at higher pressure the strongest OODR lines would become saturated. Decreasing the pressure below 50 mTorr, on the other hand, would not allow us to detect the weakest lines reported in this work. To avoid saturation of the $2\nu_3$ band we would need to reduce the pressure by a factor of 10.

The V-type dips are observed also in transitions from other (weaker) bands, provided they share the lower state with the pump (i.e., $J = 2$, symmetry F_2). Unfortunately, these transitions were outside of the probed range. We added this information on line 639:

*‘However, V-type transitions are not observed in the **present** cavity-enhanced spectra, because the absorption of the $2\nu_3$ band transitions is fully saturated **at 50 mTorr, and there were no other transitions with lower state rotational quantum number $J = 2$ within the probed spectral range.**’*

We removed the same information from the end of Supplementary information 7.

7. Discussion on the comparison between the I-N₂-cooled cell used in the previous work by the same research group (Refs. 21 & 22) and the room-temperature cavity used in this work reads somehow misleading in Sec. 3 (lines 381- 385). More accurately, the benefit is from the use of a cavity, not from "room temperature".

ANSWER: We rephrased this discussion to stress that the cavity increases the interaction length and thus allows operation at room temperature. We write on lines 418-422:

'...the use of the cavity increases by more than an order of magnitude both the absorption sensitivity (by increasing the interaction length of the probe with the sample) and frequency accuracy (by eliminating the frequency shift caused by the residual drift in the pump frequency) while allowing operation at room temperature.'

#####

We hope that we have satisfactorily addressed all Reviewers' comments and concerns. We also took the chance to make several editorial changes to further improve the clarity of our manuscript. All changes are tracked in the document appended below. The most significant changes, not mentioned above, are listed here:

1. To clarify that the two methods used in our work allow assigning only the rotational quantum number of the final state, we made the following changes:
 Line 143: 'The high SNR and frequency precision allow using two independent methods of assigning **the rotational quantum number of the** final state of the probe transitions without the need to rely on theoretical predictions.'
 Line 352: 'We assign the **branches** of the detected OODR probe transitions...'
 Line 374: 'The second method of **branch** assignment...'
 Line 493: '... assigning the **rotational quantum number** of the final states of the probe transitions'
 Line 524: '...assignment of **the rotational quantum number of the** final state...'
2. Line 153 and other: We changed 'TheoReTS' to 'TheoReTS/HITEMP' to reflect that we extract the TheoReTS data from the HITEMP database.
3. Figure 1: We changed the order of the panels because in text we refer to 1(a) before 1(b).
4. On line 272 we added the information that our work is the first demonstration of sub-Doppler spectroscopy using cavity-enhanced comb spectroscopy: 'However, none of the previously demonstrated cavity-enhanced comb-based spectrometers had **sub-Doppler resolution and** the capability to detect hot-band transitions.'
5. Line 385: '**Common** final states for different combinations of pump and probe transitions are easily identified as states whose term values agree within the experimental uncertainty, **while the separations between the different final states are significantly larger than the experimental uncertainty. This confirms that the experimental uncertainties are not underestimated.**' We added this to clarify that there were no other potential candidates for combination differences that agreed just slightly outside the experimental uncertainty.
6. Line 652: The mean ratio for the Q-pumped data is changed from 6300(300) to 6200(400) – this was a misprint before. The ratios for the R-pumped data changed after we recalculated some line intensities as described in response to comment 6 of Reviewer #1.
7. Line 802: We added acknowledgement for Michael Rey for providing the final state assignments from the non-empirical effective Hamiltonian described in his work, Ref. ³³.

8. Figure S2: We changed the beginning of the caption from '**Fits to strong and weak probe lines**. Comparison between a strong (a) and a weak (b) probe line...' to '**Fits to probe lines with high and low SNR**. Comparison of fits to probe lines with (a) **high SNR** and (b) **low SNR**...'
9. In Supplementary information 6 we added reference to the now published work of K.K. Lehmann about polarization-dependent intensity ratios in double resonance spectroscopy: '**The linear polarization-dependent intensity ratios predicted for strongly saturated inhomogeneously-broadened pump transition and unsaturated probe transition are given in Table IV of Ref. ⁶**.'

We hope that after these revisions you will find our manuscript suitable for publication.

Best regards,

Aleksandra Foltynowicz, on behalf of all authors

**Sub-Doppler optical-optical double-resonance**
**spectroscopy using a cavity-enhanced frequency**
**comb probe**

**Vinicius Silva de Oliveira¹, Isak Silander¹, Lucile Rutkowski²,**
**Grzegorz Sobon³, Ove Axner¹, Kevin K. Lehmann⁴, and Aleksandra**
**Foltynowicz^{1,*}**

¹ Department of Physics, Umeå University, 901 87 Umeå, Sweden

[revised manuscript text omitted]

~~6400(1200)~~.

**Data availability statement:** The data that support the findings of this study
are available from the corresponding author upon reasonable request.

Commented [AFM5]: This was a missprint before

Commented [AFM6]: This changed after modifying the intensities of five lines measured with only one polarization

Deleted: 6300

Deleted: 300

Deleted: 6300

Deleted: i

Deleted: 6600

Deleted: 1100

**References**

- 1 Jourdanneau, E., Chaussard, F., Saint-Loup, R., Gabard, T. & Berger,
H. The methane Raman spectrum from 1200 to 5500 cm^{-1} : A first step
toward temperature diagnostic using methane as a probe molecule in
combustion systems. *J. Mol. Spectr.* **233**, 219-230,
doi:10.1016/j.jms.2005.07.004 (2005).
- Bauke, S. *et al.* Optical sensor system for time-resolved quantification
of methane concentrations: Validation measurements in a rapid
compression machine. *J. Quant. Spectr. Rad. Transf.* **210**, 101-110,
doi:10.1016/j.jqsrt.2018.02.016 (2018).
- Koroglu, B., Neupane, S., Pryor, O., Peale, R. E. & Vasu, S. S. High
temperature infrared absorption cross sections of methane near 3.4 μm
in Ar and CO_2 mixtures. *J. Quant. Spectr. Rad. Transf.* **206**, 36-45,
doi:10.1016/j.jqsrt.2017.11.003 (2018).
- Swain, M. R., Vasisht, G. & Tinetti, G. The presence of methane in the
atmosphere of an extrasolar planet. *Nature* **452**, 329-331,
doi:10.1038/nature06823 (2008).
- Gasman, D., Min, M. & Chubb, K. L. Investigating the detectability of
hydrocarbons in exoplanet atmospheres with JWST. *Astronomy and*
*Astrophysics* **659**, A114, doi:10.1051/0004-6361/20214146 (2022).
- Miles, B. E. *et al.* The JWST Early-release Science Program for Direct
Observations of Exoplanetary Systems II: A 1 to 20 μm spectrum of the
planetary-mass companion VHS 1256–1257 b. *Astrophys. J. Lett.* **946**,
L6, doi:10.3847/2041-8213/acb04a (2023).

- Berne, O. *et al.* Formation of the methyl cation by photochemistry in a
protoplanetary disk. *Nature* **621**, 56-59, doi:10.1038/s41586-023-
06307-x (2023).
- Ulenikov, O. N. *et al.* Survey of the high resolution infrared spectrum
of methane ($^{12}\text{CH}_4$ and $^{13}\text{CH}_4$): Partial vibrational assignment extended
towards $12\,000\text{ cm}^{-1}$. *J. Chem. Phys.* **141**, doi:10.1063/1.4899263
(2014).
- Rey, M. *et al.* New accurate theoretical line lists of $^{12}\text{CH}_4$ and $^{13}\text{CH}_4$ in
the $0\text{--}13400\text{ cm}^{-1}$ range: Application to the modeling of methane
absorption in Titan's atmosphere. *Icarus* **303**, 114-130,
doi:10.1016/j.icarus.2017.12.045 (2018).
- Yurchenko, S. N., Amundsen, D. S., Tennyson, J. & Waldmann, I. P. A
hybrid line list for CH_4 and hot methane continuum. *Astronomy &*
*Astrophysics* **605**, A95, doi:10.1051/0004-6361/201731026 (2017).
- Hargreaves, R. J. *et al.* An accurate, extensive, and practical line list of
methane for the HITEMP database. *Astroph. J. Suppl. Ser.* **247**, 55,
doi:10.3847/1538-4365/ab7a1a (2020).
- Nikitin, A. V. *et al.* Improved line list of $^{12}\text{CH}_4$ in the $8850\text{--}9180\text{ cm}^{-1}$
region. *J. Quant. Spectr. Rad. Transf.* **239**, 106646,
doi:10.1016/j.jqsrt.2019.106646 (2019).
- Amyay, B. *et al.* New investigation of the ν_3 C–H stretching region of
$^{12}\text{CH}_4$ through the analysis of high temperature infrared emission
spectra. *J. Chem. Phys.* **148**, doi:10.1063/1.5023331 (2018).

- Ghysels, M. *et al.* Laser absorption spectroscopy of methane at 1000 K
near 1.7 μm : A validation test of the spectroscopic databases. *J. Quant.*
*Spectr. Rad. Transf.* **215**, 59-70, doi:10.1016/j.jqsrt.2018.04.032
(2018).
- Dudás, E. *et al.* Non-LTE spectroscopy of the tetradecad region of
methane recorded in a hypersonic flow. *Icarus* **394**, 115421,
doi:10.1016/j.icarus.2022.115421 (2023).
- de Martino, A., Frey, R. & Pradere, F. Double-resonance spectroscopy
of methane - theoretical vibration-rotation intensities and experimental
investigations of the lower ($3\nu_3$, F_2) level. *Mol. Phys.* **55**, 731-749,
doi:10.1080/00268978500101691 (1985).
- Hu, C. L. *et al.* Optical-optical double-resonance absorption
spectroscopy of molecules with kilohertz accuracy. *J. Phys. Chem. Lett.*
**11**, 7843-7848, doi:10.1021/acs.jpcelett.0c02136 (2020).
- Okubo, S., Inaba, H., Okuda, S. & Sasada, H. Frequency measurements
of the $2\nu_3A_1 - \nu_3$ band transitions of methane in comb-referenced
infrared-infrared double-resonance spectroscopy. *Phys. Rev. A* **103**,
022809, doi:10.1103/PhysRevA.103.022809 (2021).
- Nishiyama, A. *et al.* Doppler-free dual-comb spectroscopy of Rb using
optical-optical double resonance technique. *Opt. Express* **24**, 25894-
25904, doi:10.1364/oe.24.025894 (2016).
- Nishiyama, A. *et al.* Optical-optical double-resonance dual-comb
spectroscopy with pump-intensity modulation. *Opt. Express* **27**, 37003-
37011, doi:10.1364/oe.27.037003 (2019).

- Foltynowicz, A. *et al.* Sub-Doppler double-resonance spectroscopy of
methane using a frequency comb probe. *Phys. Rev. Lett.* **126**, 063001,
doi:10.1103/PhysRevLett.126.063001 (2021).
- Foltynowicz, A. *et al.* Measurement and assignment of double-
resonance transitions to the 8900-9100-cm⁻¹ levels of methane. *Phys.*
*Rev. A* **103**, 022810, doi:10.1103/PhysRevA.103.022810 (2021).
- Foltynowicz, A., Maslowski, P., Fleisher, A. J., Bjork, B. J. & Ye, J.
Cavity-enhanced optical frequency comb spectroscopy in the mid-
infrared - application to trace detection of hydrogen peroxide. *Appl.*
*Phys. B* **110**, 163-175, doi:10.1007/s00340-012-5024-7 (2013).
- Foltynowicz, A., Ban, T., Maslowski, P., Adler, F. & Ye, J. Quantum-
noise-limited optical frequency comb spectroscopy. *Phys. Rev. Lett.*
**107**, 233002, doi:10.1103/PhysRevLett.107.233002 (2011).
- Maslowski, P. *et al.* Surpassing the path-limited resolution of Fourier-
transform spectrometry with frequency combs. *Phys. Rev. A* **93**,
021802(R), doi:10.1103/PhysRevA.93.021802 (2016).
- Rutkowski, L., Maslowski, P., Johansson, A. C., Khodabakhsh, A. &
Foltynowicz, A. Optical frequency comb Fourier transform
spectroscopy with sub-nominal resolution and precision beyond the
Voigt profile. *J. Quant. Spectr. Rad. Transf.* **204**, 63-73,
doi:10.1016/j.jqsrt.2017.09.001 (2018).
- Cole, R. K., Makowiecki, A. S., Hoghooghi, N. & Rieker, G. B.
Baseline-free quantitative absorption spectroscopy based on cepstral

analysis. *Opt. Express* **27**, 37920-37939, doi:10.1364/oe.27.037920
(2019).

Hoghooghi, N. *et al.* Broadband coherent cavity-enhanced dual-comb
spectroscopy. *Optica* **6**, 28-33, doi:10.1364/OPTICA.6.000028 (2019).

Fleisher, A. J. *et al.* Mid-infrared time-resolved frequency comb
spectroscopy of transient free radicals. *J. Phys. Chem. Lett.* **5**, 2241-
2246, doi:10.1021/jz5008559 (2014).

Lyulin, O. M. *et al.* Measurements of self-broadening and self-pressure-
induced shift parameters of the methane spectral lines in the 5556–
6166cm⁻¹ range. *J. Quant. Spectr. Rad. Transf.* **112**, 531-539,
doi:10.1016/j.jqsrt.2010.10.010 (2011).

Okubo, S., Nakayama, H., Iwakuni, K., Inaba, H. & Sasada, H.
Absolute frequency list of the ν₃-band transitions of methane at a
relative uncertainty level of 10⁻¹¹. *Opt. Express* **19**, 23878-23888,
doi:10.1364/oe.19.023878 (2011).

Lehmann, K. K. Polarization-dependent intensity ratios in double
resonance spectroscopy. *arXiv:2308.09828*,
doi:10.48550/arXiv.2308.09828 (2023).

Rey, M. Novel methodology for systematically constructing global
effective models from ab initio-based surfaces: A new insight into high-
resolution molecular spectra analysis. *J. Chem. Phys.* **156**, 224103,
doi:10.1063/5.0089097 (2022).

Private communication with Hiroyuki Sasada.

- Abe, M., Iwakuni, K., Okubo, S. & Sasada, H. Accurate transition
frequency list of the n_3 band of methane from sub-Doppler resolution
comb-referenced spectroscopy. *J. Opt. Soc. Am. B* **30**, 1027-1035,
doi:10.1364/josab.30.001027 (2013).
- Nikitin, A. V., Rey, M. & Tyuterev, V. G. Accurate line intensities of
methane from first-principles calculations. *J. Quant. Spectr. Rad.*
*Transf.* **200**, 90-99, doi:10.1016/j.jqsrt.2017.05.023 (2017).
- Sobon, G. *et al.* All-in-fiber amplification and compression of coherent
frequency-shifted solitons tunable in the 1800-2000 nm range. *Phot.*
*Res.* **6**, 368-372, doi:10.1364/prj.6.000368 (2018).
- Khodabakhsh, A., Johansson, A. C. & Foltynowicz, A. Noise-immune
cavity-enhanced optical frequency comb spectroscopy: a sensitive
technique for high-resolution broadband molecular detection. *Appl.*
*Phys. B* **119**, 87-96, doi:10.1007/s00340-015-6010-7 (2015).

**Acknowledgments:** The authors thank Hiroyuki Sasada for providing the
ground state term value from his unpublished work, and Michael Rey for
providing the final state assignments from the non-empirical effective
Hamiltonian described in his work, Ref. ³³.

Deleted: upper state

**Author contributions:** K.K.L. and A.F. conceived the idea. V.S.O. and
I.S. implemented the experiment and performed the measurements. V.S.O
analyzed and visualized the data. L.R. and K.K.L. contributed theoretical
predictions and analysis tools. A.F., G.S., and O.A. provided resources.
810 A.F. supervised the project and wrote the manuscript. V.S.O, L.R., G.S.,
O.A, and K.K.L. revised the manuscript.

**Competing interest statement:** The authors declare no competing
interests.

**Funding:** A.F. acknowledges the Knut and Alice Wallenberg Foundation
(KAW 2015.0159, KAW 2020.0303) and the Swedish Research Council
(2020-00238). L.R. acknowledges the French National Research Agency
(ANR-19-CE30-0038-01); G.S. acknowledges the Foundation for Polish
Science (POIR.04.04.00-00-434D/17-00); O.A. acknowledges the Kempe
Foundation (JCK 1317.1), K.K.L. acknowledges the National Science
Foundation (grant: CHE- 2108458).

**Sub-Doppler optical-optical double-resonance
spectroscopy using a cavity-enhanced frequency
comb probe: Supplementary information**

**Vinicius Silva de Oliveira¹, Isak Silander¹, Lucile Rutkowski²,
Grzegorz Soboń³, Ove Axner¹, Kevin K. Lehmann⁴, and Aleksandra
Foltynowicz^{1,*}**

¹ Department of Physics, Umeå University, 901 87 Umeå, Sweden

² Univ Rennes, CNRS, IPR (Institut de Physique de Rennes)-UMR 6251,
F-35000 Rennes, France

³ Faculty of Electronics, Photonics and Microsystems, Wrocław University
of Science and Technology, Wybrzeże Wyspiańskiego 27, 50-370
Wrocław, Poland

⁴ Departments of Chemistry & Physics, University of Virginia,
Charlottesville, VA 22904, USA

Corresponding author: aleksandra.foltynowicz@umu.se

1. Cavity finesse

The cavity finesse was evaluated from a measurement of the cavity ring-down time at 11 points between 5910 and 5980 cm^{-1} . The mean of 100 retrieved finesse values at each wavenumber is shown by the black markers in Fig. S1, where the error bars are standard deviations of the mean. A linear fit to the finesse (red curve) was used to fix the finesse values in the model for line fitting. The design mirror wavelength (with maximum reflectivity) was 1580 nm (6300 cm^{-1}), and the linear fit agrees well with the mirror reflectivity data. The 1σ confidence interval of the fit (shaded grey) indicates 3% relative uncertainty of the finesse, which was propagated to the retrieved integrated absorption values.

Figure S1. **Cavity finesse.** The cavity finesse evaluated from the cavity ring-down time (black markers) together with a linear fit to the data (red curve), the 1σ confidence interval (shaded) and the fit residuals (lower window).

2. Data averaging

During the acquisition, the stepping of the repetition rate was performed in two different ways. For measurements with the pump locked to the R(2, F_2) transition, 5 pairs of sample and background interferograms were taken at each f_{rep} value. The f_{rep} was then stepped using the procedure described in the main paper. For the parallel relative pump/probe polarizations, this process was repeated 9 times to acquire a total of 45 interferograms at each f_{rep} value. For measurements with the pump locked to the P(2, F_2) and Q(2, F_2) transitions, one pair of sample and background interferograms was measured, then the f_{rep} was stepped, and a new pair was measured. This scan was repeated 5 times to yield 5 averages. The P(2, F_2)-pumped dataset was measured twice, with different Pound-Drever-Hall (PDH) comb-cavity locking points, to optimize the SNR in the spectral region where probe transitions are found.

The f_{ceo} was indirectly stabilized via the two-point PDH lock¹ and monitored using a counter with 1 s integration time. We found that during the long f_{rep} scans the f_{ceo} drifted by up to 200 kHz, which we attribute to drifts in the offsets of the PDH locks. This implied that the frequency axes in the five consecutive P(2, F_2)- and Q(2, F_2)-pumped spectra were known accurately but were not identical. Therefore, instead of averaging the 5 spectra, we combined them into one interleaved spectrum (*i.e.*, having 5 points at each f_{rep} value). The measurement with the pump locked to the R(2, F_2) transition could be averaged 5 times, since we acquired 5 consecutive sample-background pairs at each f_{rep} step, during which the

Deleted: i

Deleted: as

f_{ceo} was constant. However, for consistency, for the fitting, we combined the R(2, F_2)-pumped spectra in the same way as the P(2, F_2)- and Q(2, F_2)-pumped spectra. The weakest lines in the R(2, F_2)-pumped spectrum could be observed only after averaging all 45 spectra from the 9 scans of f_{rep} . To average the spectra from the different f_{rep} scans that had slightly different f_{ceo} values, we interpolated the data to the same frequency axis, *i.e.*, the same f_{ceo} value.

3. Line fitting

To retrieve the parameters of probe transitions, we fit the cavity transmission model¹, where the line shape was assumed to comprise a sum of a narrow and strong Lorentzian function for the sub-Doppler probe transition, and a wider and weaker Gaussian function for the Doppler-broadened background absorption originating from thermal redistribution of the population of the upper pump level by elastic velocity changing collisions². The fitting range was ± 500 MHz, which is more than 3 times the width of the Doppler-broadened background. Figure S2 shows examples of fits (zoomed to ± 250 MHz for clarity) to two probe lines in the $R(2, F_2)$ -pumped spectrum with each type of contribution indicated separately. The fit model was implemented in Matlab and fitted using the trust-region nonlinear least squares algorithm. The fit parameters were the integrated absorptions and widths of the Lorentzian and Gaussian profiles, the center frequency common for the Lorentzian and Gaussian profiles, and the wavelength-dependent comb-cavity phase offset. Dispersion in the cavity mirror coatings causes a non-zero comb-cavity resonance offset away from the PDH locking points. The cavity-transmission function also contains molecular dispersion, which results in an additional shift of the cavity resonances with respect to the comb lines. This shift has a dispersive shape as a comb mode is moved across an absorption line.¹ Since this molecular contribution to the shift is small compared to the width of the cavity modes, the change in cavity transmission is approximately linear with this molecular induced shift. This effect is well

Deleted: wider than

Deleted: cavity dispersion, where the latter was modeled via a

understood and included in the model, and it does not shift the fitted center frequency. The cavity finesse was fixed to the values from the fit to the experimental data (see Section 1). For the weak lines, the Gaussian width was fixed to 141.7 MHz, while the ratio of the sub-Doppler and Doppler-broadened integrated absorptions was fixed to 0.72, based on the mean of the values from fits to strong lines. The sidebands originating from the frequency modulation used in the Lamb dip stabilization of the pump that appear at ± 120 MHz around the central peak (twice the modulation frequency due to the probe being twice the pump frequency) were excluded from the fit and are marked in gray in Fig. S2. For probe lines overlapping with Doppler-broadened transitions from the $2\nu_3$ band, the absorption and dispersion of the $2\nu_3$ band lines (based on the parameters from the HITRAN database³) were included in the model of the background spectrum, as their presence modifies the cavity losses, and thus the enhancement, even though they are removed by normalization.

The structure visible in the residuals around the line centers indicates that the model based on a sum of a single Lorentzian and single Gaussian function is not fully sufficient to describe the observed line shape.

Attempts to include a sum of individual Lorentzian functions for the different M_J components, where M_J is the quantum number for the projection of total angular momentum on the axis defined by the pump electric field, did not yield an improvement. More accurate modeling of the observed line shape will be a subject of further study. For now, we note that the residuals are symmetric around the line center, and thus the

inaccuracy in the model does not affect the accuracy of the center frequency retrieval.

We note that in Fig. S2 all measurement points are combined (interleaved) as described in Section 2, whereas in Fig. 2 in the main paper the 5 spectra are averaged for clarity.

Figure S2: Fits to probe lines with high and low SNR. Comparison of fits to probe lines with (a) high SNR and (b) low SNR, detected in the spectrum with the pump locked to the ν_3 $R(2, F_2)$ transition with parallel relative pump/probe polarizations. The 5 combined spectra are shown in black in the upper windows. The green curves show the fit of the model, while the blue and magenta curves display the Lorentzian and Gaussian parts, respectively. The lower windows show residuals of the fits. The ranges around the FM sidebands that are excluded from fitting are indicated in gray.

Deleted: strong and weak

Deleted: between a

Deleted: strong

Deleted:

Deleted: a weak

Deleted: probe line

4. Frequency uncertainty

4.1 Frequency scale calibration

The frequency scale was calibrated using the sub-nominal resolution method of Refs. ^{4,5}, where precise sampling of the comb modes is obtained by setting the nominal resolution of the Fourier transform spectrometer (FTS) equal to comb mode spacing and shifting the origin of the FTS frequency axis to account for the f_{ceo} . The fine calibration of the frequency axis was performed by minimizing the instrumental line shape on the strongest sub-Doppler probe line in each spectrum. To do that, we generated spectra of the strongest line using a set of values of the reference laser wavelength (which calibrates the optical path difference, and thus the nominal resolution) differing by ~ 10 fm. Afterward, we performed fits to the resulting spectra, as described in Section 3, and plotted the standard deviation of the fit residuals as a function of the reference laser wavelength. The minimum of the standard deviation corresponds to the lowest instrumental line shape and thus to the best match between the comb mode spacing and the FTS nominal resolution. From the depth of this minimum, we estimated the uncertainty in the reference laser wavelength to be 50 fm. Finally, we fit a line to the center frequencies from fits to those spectra plotted as a function of the reference laser wavelength. From the slope of this line and the 50 fm uncertainty of the reference laser wavelength, we obtained an uncertainty of the center frequency of 30 kHz ($1 \times 10^{-6} \text{ cm}^{-1}$), which is comparable to the fit

uncertainty for the strongest lines, but negligible compared to the total uncertainty (see Section Frequency uncertainty in the main paper and 4.2 below).

Deleted: 2.3

We note that in Ref. ⁵ it was shown that when the repetition rate of the comb is much larger than the line width of the transition (as is the case here for the sub-Doppler probe lines) the center frequency should not change with reference laser wavelength calibration. However, here, the sub-Doppler lines reside on top of a Doppler-broadened component, whose width is of the same order as f_{rep} . In the fit, the center frequency is assumed equal for both components, therefore, the center frequency depends slightly on the reference laser wavelength calibration.

4.2 Frequency uncertainty model

Deleted: accuracy

As discussed in the main paper, to investigate the long-term stability of the center frequency, we performed fits to 9 probe transitions detected in the 45 spectra recorded over 12 h with the pump locked to the ν_3 R(2, F_2) transition and parallel relative pump/probe polarization. The analysis of the center frequencies from the 45 individual fits to each line revealed that their spread is larger than the uncertainty of the individual fits, which we attribute to a residual uncorrected baseline drift.

Deleted: accuracy

Deleted: determination

Figure S3 shows the 1σ standard deviation of the center frequencies from the long-term measurement (green markers, left axis) and the fit precision of all probe lines detected in the five combined P(2, F_2), Q(2, F_2), and R(2, F_2)-pumped spectra (blue, red, and black markers, respectively), as a function of the SNR of the line in a single measurement.

The SNR was calculated as the ratio of the peak absorption (1-transmission) and the standard deviation of the residuals (excluding the FM sidebands and the line center due to the mismatch between the model and the measurement observed for high-SNR lines). For the long-term series (green markers), the SNR was taken as the mean SNR of the 45 consecutive spectra, where the horizontal error bar is the standard deviation of the mean. For the probe lines fitted in the five combined spectra (blue, red, and black markers, respectively, right axis), the standard deviation of the residuals corresponds to the noise in a single measurement rather than an average of 5 measurements, since the data are interleaved rather than averaged (as described in Section 2).

Deleted: [%]

Deleted: the individual fits, we note that since the data is interleaved (as described in Section 2),

Deleted: from a fit to the 5 combined spectra

Deleted: one average

Deleted: .

Since the uncertainty from the long-term measurement is systematically larger than the fit precision for all lines, we developed a model for the frequency uncertainty based on the SNR dependence of the standard deviation of the long-term results. The blue curve in Fig. S3 shows a fit of a model function $U = (a^2 + b^2 / \text{SNR}^2)^{1/2}$ to the observed relation between the uncertainty and the SNR from the 45-measurement series, where a and b are fitting parameters. We used this function to estimate the center frequency uncertainty for all detected lines based on their single-measurement SNR evaluated as described above. Since the baseline fluctuations, to which we attribute the increased frequency uncertainty, do not depend on the choice of the pump transition, we applied the model also to the lines that were not measured 45 times, *i.e.*, the P(2, F_2)- and Q(2,

Deleted: [especially for

F_2)-pumped lines, and the 6 weakest $R(2, F_2)$ -pumped lines visible only in the spectrum averaged 45 times.

Figure S3. **Frequency uncertainty model.** Center frequency uncertainty as a function of the SNR of the individual lines in a single measurement. The green markers (left vertical axis) show the standard deviations of the center wavenumbers, evaluated from fits to the long-term measurements with the pump locked to the $R(2, F_2)$ transition. The blue, red, and black markers (right vertical axis) show center frequency fit precision for all lines detected with the pump on the $P(2, F_2)$, $Q(2, F_2)$, and $R(2, F_2)$ transitions, respectively.

Deleted: accuracy

Deleted: uncertainties

5. Comparison to previous measurements

Two probe transitions in the spectrum measured with the pump locked to the $v_3 P(2, F_2)$ transition were observed in our previous work², where the sample was contained in the liquid-nitrogen-cooled single-pass cell. Figure S4 shows one of these transitions measured in the cell (from Ref. ², 16 averages, 40 mTorr, 110 K, measurement with pump beam on only) and in the cavity (this work, 1 average, 296 K, 50 mTorr, ratio of spectra with pump on and off). The SNR in the cell was 10 after 3.2 h of averaging, while in the cavity it is 200 in 16.7 minutes, *i.e.*, less than 1/10 of the time.

Normalized to the same acquisition time, this implies that the SNR in the cavity is a factor of $200 \times 10^{-1/2} = 60$ better than in the cell. This improvement is smaller than the factor of 700 improvement in the noise equivalent absorption sensitivity (see Section Sensitivity in the main manuscript) because the pump absorption was stronger in the cooled cell.

The temperature dependence of the absorption coefficient of the pump transitions at a constant pressure is given by $(T_2/T_1)^{-3} \times \exp[-(c_2 E_{\text{rot}})(1/T_2 - 1/T_1)]$, where the T^{-3} dependence is a product of the temperature dependence of the number density (T^{-1}), the inverse of the partition function ($T^{-3/2}$ for a nonlinear molecule), and the normalized line shape function ($T^{-1/2}$ for Doppler-broadened transitions), and the second term represents the ratio of the Boltzmann factors, where E_{rot} is the lower pump level term value and c_2 is the second radiation constant. For $T_1 = 296$ K, $T_2 = 110$ K, and $E_{\text{rot}} = 31.4423878$ cm^{-1} , this yields a factor of 14.6.

Considering also the difference in sample pressure (50 mTorr vs 30

Deleted: cavity

Deleted: not equal to

Deleted: 2.2

Deleted: signal

Deleted: is

mTorr), the absorption coefficient of the pump transitions was 11.7 stronger in the cell than in the cavity. This implies that the improvement in the SNR can be estimated to $700/11.7 = 60$, which agrees with what is observed in the data. We note that in the main paper we state the improvement of $700/14.6 = 50$, which is valid under same pressure conditions.

A comparison of the wavenumbers and upper state term values of the two lines detected both in the cell and in the cavity is shown in Table S1, demonstrating a more than 1 order of magnitude improvement in frequency accuracy. This improvement is confirmed by the fact that all final states in the combination differences agree within their uncertainties.

Figure S4. **Sensitivity comparison.** The R(1) probe line detected in the spectrum measured with the pump locked to the ν_3 P(2, F_2) transition in (a) the single pass cell (from Ref. ²) and (b) in the enhancement cavity in this work.

Table S1: **Comparison between probe transition wavenumbers and final state term values** for two OODR probe lines detected both in the cell² and in the cavity (this work).

Pump transition, ν_3 band	Probe transition	Probe transition wavenumber [cm ⁻¹]	Final term state value [cm ⁻¹]	Ref.
P(2, F ₂)	R(1)	5979.04297(5)	9009.47939(5)	²
		5979.042972(4)	9009.479391(4)	This work
P(2, F ₂)	R(1)	5948.26760(5)	8979.70402(5)	²
		5948.267590(3)	8979.704010(3)	This work

Deleted: erence

6. Predicted and experimental polarization-dependent intensity ratios

Deleted: P

As explained in Ref. ⁶, the polarization-dependent intensity ratios can be predicted as the ratio of the sum over M_J quantum states of the products of the absolute value of the transition dipole for the pump transition times the square of the transition dipole for the probe transition evaluated for each relative polarization. The reason this ratio is proportional to the pump transition dipole rather than its square is that in the strong pumping limit, the steady-state population integrated over detuning is proportional to the Rabi frequency. The linear polarization-dependent intensity ratios predicted for strongly saturated inhomogeneously-broadened pump transition and unsaturated probe transition are given in Table IV of Ref. ⁶.

Deleted: T

Deleted:

The predicted intensity ratios for all pump-probe combinations observed in this work are listed in the second last column of Table II in the main manuscript.

Deleted: A publication detailing the calculations is under preparation. ...

Deleted: observed

The accuracy of the measured polarization-dependent intensity ratios was limited by the quality of the probe polarization. This polarization was slightly elliptical because of the propagation in the nonlinear fiber and the fiberized optical circulator. 4.2% of the probe power in front of the cavity was along the minor axis, which corresponds to an ellipticity of 0.2. Compared to that, the pump polarization was purely linear. To calculate the uncertainties in the measured intensities and their ratios caused by the ellipticity of the probe polarization, we assumed that the polarizations of

the pump and probe are linear but offset by an angle $\varphi = 0.2$ from being perfectly parallel or perpendicular.

For the integrated intensities measured with parallel and perpendicular relative pump/probe polarizations, I_{\parallel} and I_{\perp} , we assumed a relative error of $\sin^2(\varphi) = 4.2\%$ caused by the ellipticity of the probe light.

To evaluate the uncertainty in the measured polarization-dependent intensity ratios, we compared the intensity ratios predicted for perfectly parallel and perpendicular pump and probe polarizations, given by

$$R_{\parallel/\perp}(0) = \frac{I_{\parallel}}{I_{\perp}} \quad (1)$$

with ratios predicted for an angle φ , given by

$$R_{\parallel/\perp}(\varphi) = \frac{I_{\perp} \sin^2(\varphi) + I_{\parallel} \cos^2(\varphi)}{I_{\perp} \cos^2(\varphi) + I_{\parallel} \sin^2(\varphi)} = \frac{R_{\parallel/\perp}(0) + \sin^2(\varphi)[1 - R_{\parallel/\perp}(0)]}{1 - \sin^2(\varphi)[1 - R_{\parallel/\perp}(0)]} \quad (2)$$

We then took twice the difference $2|R_{\parallel/\perp}(\varphi) - R_{\parallel/\perp}(0)|$, given by

$$2|R_{\parallel/\perp}(\varphi) - R_{\parallel/\perp}(0)| = \frac{2\sin^2(\varphi)[1 - R_{\parallel/\perp}^2(0)]}{1 - \sin^2(\varphi)[1 - R_{\parallel/\perp}(0)]} \quad (3)$$

as the uncertainty in the measured polarization-dependent intensity ratio introduced by the probe ellipticity φ . We note that this uncertainty is smallest for ratios closest to 1, and larger for ratios that are farther from 1.

7. Results for all probe lines

Table S2 contains the results of the fits to all detected probe lines. Column 1 shows the pump transition in the ν_3 band. Column 2 lists the experimental probe transition assignment (in agreement with TheoReTS/HITEMP⁷). The 6 weakest transitions, marked by an asterisk, were detected only in the spectrum averaged 45 times with parallel pump/probe polarizations, and their assignments (if available) are from TheoReTS/HITEMP only. Column 3 lists the measured center wavenumber, calculated as a weighted mean of the wavenumbers found from fits to spectra recorded with the two pump polarizations, where the weight was the inverse of the square of the uncertainty calculated from the SNR-based model (see Section 4.2). Column 4 shows the final state term value, calculated as described in Section Line assignments in the main paper. Column 5 states the integrated absorption of the probe transition, calculated as a weighted mean of the absorption measured with parallel and perpendicular relative pump/probe polarizations, with weight 1 for parallel and 2 for perpendicular polarization, which yields an isotropic average independent of the value of the M_J quantum number. For the 6 weakest lines that were measured only with parallel pump/probe polarization, we recalculated the measured absorption to the weighted mean using the predicted polarization-dependent intensity ratios. The uncertainty is a combination of the fit uncertainty, the uncertainty in the finesse determination (3%), and polarization of the probe (4.2%). Column 6 provides the ratio of the line intensities measured with the parallel and

Deleted: is

Deleted: 2.4

Deleted: list the integrated absorption of the measurement with parallel pump/probe polarization only

Deleted: and

perpendicular relative pump/probe polarizations. The uncertainty is a combination of the fit uncertainty of the two integrated absorptions, and the uncertainty caused by the ellipticity of the probe given by Eq. (3) above. The uncertainty in the finesse is not taken into account, since the contribution from the finesse cancels when taking the ratio of the two intensities. Column 7 shows the half width at half maximum of the probe transition, calculated as a weighted mean of the widths found from fits to spectra recorded with the two pump polarizations, where the weight was the inverse of the fit variance. Columns 8 and 9 display the integrated absorption and the width of the Gaussian part, respectively. For the weaker lines, the ratio of the sub-Doppler and the Doppler-broadened integrated absorptions was fixed to 0.72, i.e. the mean value found from the fits to the strongest lines. Similarly, the width of the Doppler-broadened contribution was fixed to 141.7 MHz, and it is given without uncertainty in column 9. This width is slightly smaller than the 165 MHz thermal Doppler width at 296 K. Columns 10 and 11 show the predicted probe transition wavenumber from TheoReTS/HITEMP⁷ and its difference with respect to the observed wavenumber [plotted in Fig. 6(c) in the main paper]. Column 12 shows the final state dominant assignment obtained from the Hamiltonian described in Ref. ⁸. Columns 13 and 14 give the line intensity from TheoReTS/HITEMP at 296 K⁷ and the integrated absorption calculated at the experimental conditions of 296 K and 50 mTorr.

Deleted: limited

Deleted: (3)3

Deleted: accuracy with which the relative pump/probe polarization is set using the half-wave plate

Deleted: The HWHM width of the sub-Doppler probe lines is, on average, 5 MHz, limited primarily by pump power broadening.

Deleted: is

Deleted: (the mean value from the fits to the strongest lines)

Deleted: less

Deleted: For these lines, the ratio of the sub-Doppler and the Doppler-broadened integrated absorptions was fixed to 0.72, the mean ratio found from fits to strongest lines.

Deleted: 2

Deleted: 3

Deleted: In principle, the absolute line strength of the OODR probe transitions could be estimated from the integrated strength of the Bennet hole observed in the V-type transitions with known line strength². However, with the cavity, the V-type transitions in the probed range were too strongly saturated to use this method.[¶]

Table S2: Results of fits to all lines. See description in text.

1	2	3	4	5	6	7	8	9	10	11	12	13	14
Pump transition in the ν_3 band	Probe transition	Probe transition wavenumber [cm ⁻¹]	Final state term value [cm ⁻¹]	Probe transition integrated absorption [10 ⁻⁹ cm ⁻²]	Polar. depend. intensity ratio	Probe transition width [MHz]	Gaussian integrated absorption [10 ⁻⁹ cm ⁻²]	Gaussian width [MHz]	TheoReTS transition wavenumber [cm ⁻¹]	Obs. - pred. wavenum. [cm ⁻¹]	Final state assignment from Ref. ⁸	TheoReTS transition intensity [10 ⁻³⁰ cm/molec]	TheoReTS integrated absorption [10 ⁻¹³ cm ²]
P(2,F ₂)	R(1)	5948.267590(3)	8978.704010(3)	2.26(9)	0.91(5)	5.093(8)	1.50(3)	144.3(4)	5947.67695	0.59	3v ₃ (F ₁)	220.40	3.59
	R(1)	5964.06227(2)	8994.49869(2)	0.081(4)	1.01(8)	6.5(3)	0.058(2)	141.7	5965.37197	-1.31	5v ₂ +3v ₄ (F ₁)	7.79	0.13
	R(1)	5979.042972(3)	9009.479392(3)	0.56(2)	0.98(6)	4.459(8)	0.396(9)	125.5(6)	5979.71721	-0.67	v ₁ +4v ₂ (A ₁)	68.70	1.12
Q(2,F ₂)	Q(2,F ₁)	5928.61142(2)	8978.70401(2)	0.56(4)	1.5(3)	6.0(1)	0.40(2)	141.7	5928.02556	0.59	3v ₃ (F ₁)	53.28	0.87
	Q(2,F ₁)	5944.40608(2)	8994.49868(2)	0.103(5)	1.5(3)	4.31(9)	0.074(2)	141.7	5945.72057	-1.31	5v ₂ +3v ₄ (F ₁)	10.83	0.18
	R(2,F ₁)	5958.673574(6)	9008.766169(6)	2.38(10)	0.83(7)	4.73(8)	2.20(8)	143(4)	5957.83620	0.84	3v ₃ (F ₁)	243.60	3.97
	Q(2,F ₁)	5959.386797(5)	9009.479392(5)	0.89(3)	1.8(3)	4.29(1)	0.64(1)	141.7	5960.06582	-0.68	v ₁ +4v ₂ (A ₁)	82.82	1.35
R(2,F ₂)	R(3,F ₁)	5913.18732(2)	8992.78303(2)	0.062(4)	1.1(1)	3.6(3)	0.045(3)	141.7	5913.97472	-0.79	3v ₂ +v ₃ +v ₄ (F ₂)	6.48	0.11
	R(3,F ₁)	5918.14141(1)	8997.73712(1)	0.096(5)	1.05(10)	4.1(1)	0.069(3)	141.7	5918.79890	-0.66	3v ₂ +v ₃ +v ₄ (F ₂)	11.14	0.18
	*R(3,F ₁)	5920.775507(9)	9000.371213(9)	0.039(2)	==	5.67(6)	0.034(1)	141.7	5921.56979	-0.79	3v ₂ +v ₃ +v ₄ (F ₂)	4.43	0.07
	* ==	5922.51469(4)	9002.11039(4)	0.012(2)	==	4(1)	0.010(2)	141.7	==	==	==	==	==
	*R(3,F ₁)	5923.32503(5)	9002.92074(5)	0.039(6)	==	9(2)	0.033(4)	141.7	5923.25121	0.074	==	2.17	0.04
	R(3,F ₁)	5923.94848(1)	9003.54418(1)	0.44(2)	1.19(10)	5.2(2)	0.32(1)	141.7	5924.39254	-0.44	3v ₂ +v ₃ +v ₄ (F ₁)	49.03	0.80
	R(3,F ₁)	5924.26536(2)	9003.86107(2)	0.175(7)	1.1(1)	3.79(8)	0.126(3)	141.7	5924.10232	0.16	3v ₂ +v ₃ +v ₄ (F ₂)	13.00	0.21
	*R(3,F ₁)	5924.99660(1)	9004.59230(1)	0.049(4)	==	4.3(3)	0.041(2)	141.7	5924.99972	-0.003	3v ₂ +v ₃ +v ₄ (F ₂)	2.23	0.04
	* ==	5929.14547(1)	9008.74117(1)	0.112(3)	==	5.67(9)	0.036(1)	141.7	==	==	==	==	==
	Q(3,F ₁)	5929.170466(3)	9008.766172(3)	4.7(2)	0.4(1)	5.87(1)	2.81(7)	142.4(3)	5928.34007	0.83	3v ₃ (F ₁)	418.90	6.83
	P(3,F ₁)	5929.883687(4)	9009.479393(4)	0.85(3)	1.6(2)	4.51(2)	0.61(1)	141.7	5930.56969	-0.69	v ₁ +4v ₂ (A ₁)	71.45	1.16
	R(3,F ₁)	5932.279186(9)	9011.874892(9)	0.100(4)	1.2(1)	4.00(6)	0.072(2)	141.7	5932.52779	-0.25	3v ₂ +v ₃ +v ₄ (F ₂)	8.00	0.13
	*R(3,F ₁)	5932.876751(8)	9012.472457(8)	0.025(2)	==	4.6(2)	0.0214(9)	141.7	5932.88420	-0.007	3v ₂ +v ₃ +v ₄ (F ₂)	3.30	0.05
R(3,F ₁)	5935.245195(3)	9014.840901(3)	0.74(3)	1.36(9)	4.409(8)	0.64(1)	126.3(5)	5935.38477	-0.14	3v ₂ +v ₃ +v ₄ (F ₂)	58.52	0.95	

- Deleted: 2
- Deleted: 3
- Deleted: 5
- Deleted: 4
- Deleted: 3
- Deleted: 7
- Deleted: 1
- Deleted: 4
- Deleted: 3
- Deleted: 0...10
- Deleted: 3
- Deleted: 5...6
- Deleted: 6
- Deleted: 5
- Deleted: 2
- Deleted: 4...37
- Deleted: 4
- Deleted: 46...65
- Deleted: 6
- Deleted: 5
- Deleted: 0...8
- Deleted: 58...3
- Deleted: 05...120...2
- Deleted: 1
- Deleted: 4...17
- Deleted: 2
- Deleted: 1...4
- Deleted: 3
- Deleted: 1...9
- Deleted: 030...25(1)
- Deleted: 2
- Deleted: 5

References

- Foltynowicz, A., Maslowski, P., Fleisher, A. J., Bjork, B. J. & Ye, J. Cavity-enhanced optical frequency comb spectroscopy in the mid-infrared - application to trace detection of hydrogen peroxide. *Appl. Phys. B* **110**, 163-175, doi:10.1007/s00340-012-5024-7 (2013).
- Foltynowicz, A. *et al.* Measurement and assignment of double-resonance transitions to the 8900-9100-cm⁻¹ levels of methane. *Phys. Rev. A* **103**, 022810, doi:10.1103/PhysRevA.103.022810 (2021).
- Gordon, I. E. *et al.* The HITRAN2020 molecular spectroscopic database. *J. Quant. Spectr. Rad. Transf.* **277**, 107949, doi:10.1016/j.jqsrt.2021.107949 (2022).
- Maslowski, P. *et al.* Surpassing the path-limited resolution of Fourier-transform spectrometry with frequency combs. *Phys. Rev. A* **93**, 021802(R), doi:10.1103/PhysRevA.93.021802 (2016).
- Rutkowski, L., Maslowski, P., Johansson, A. C., Khodabakhsh, A. & Foltynowicz, A. Optical frequency comb Fourier transform spectroscopy with sub-nominal resolution and precision beyond the Voigt profile. *J. Quant. Spectr. Rad. Transf.* **204**, 63-73, doi:10.1016/j.jqsrt.2017.09.001 (2018).
- Lehmann, K. K. Polarization-dependent intensity ratios in double resonance spectroscopy. *arXiv:2308.09828*, doi:10.48550/arXiv.2308.09828 (2023).

- Hargreaves, R. J. *et al.* An accurate, extensive, and practical line list of methane for the HITEMP database. *Astroph. J. Suppl. Ser.* **247**, 55, doi:10.3847/1538-4365/ab7a1a (2020).
- Rey, M. Novel methodology for systematically constructing global effective models from ab initio-based surfaces: A new insight into high-resolution molecular spectra analysis. *J. Chem. Phys.* **156**, 224103, doi:10.1063/5.0089097 (2022).

REVIEWERS' COMMENTS

Reviewer #1 (Remarks to the Author):

I thank the authors for a thorough and careful response. All comments were addressed sufficiently and I support publication.

Reviewer #2 (Remarks to the Author):

I appreciate authors' efforts for the response. In term of innovation of this paper, which I was concerned about in the first review, I think that the authors' response has provided some convincing answers. Although there is no major innovation from a technical point of view, it is true that this paper has a strong impact on the reader as an application of comb-based cavity enhanced FTS to sub-Doppler spectroscopy that goes beyond the level of proof of principle, as the authors claimed.

Regarding the center frequency accuracy mentioned in my comment 2, in the first version of the manuscript, it was difficult for me to understand and led to misunderstandings. But in the revised version, their intention to use the "frequency uncertainty model" became clear. The fitting error is the dominant uncertainty in the region of poor SNR ($SNR < 50$), and the baseline drift limits the uncertainty in the region of good SNR ($SNR > 50$). It would be an appropriate method to evaluate the uncertainty of the center frequency for these measurements. Other responses, including the responses to questions from other reviewers, are also clear and convincing.

I believe that the revised manuscript is well written and ready for publication. I can recommend for publication in Nature communication.

Reviewer #3 (Remarks to the Author):

The manuscript has been adequately revised and can be accepted for publication in NC.

Dear Reviewers,

We are pleased to see that you found our responses convincing and satisfactory and that you recommend our manuscript for publication. We are happy we could clarify the points raised by Reviewer #2. Again, we thank you all for your comments that helped us improve the quality of our manuscript.

The final Reviewer comments are pasted below. We believe they do not require any response.

Best regards,

Aleksandra Foltynowicz, on behalf of all authors

#####

Reviewer #1 (Remarks to the Author):

I thank the authors for a thorough and careful response. All comments were addressed sufficiently and I support publication.

#####

Reviewer #2 (Remarks to the Author):

I appreciate authors' efforts for the response. In term of innovation of this paper, which I was concerned about in the first review, I think that the authors' response has provided some convincing answers. Although there is no major innovation from a technical point of view, it is true that this paper has a strong impact on the reader as an application of comb-based cavity enhanced FTS to sub-Doppler spectroscopy that goes beyond the level of proof of principle, as the authors claimed. Regarding the center frequency accuracy mentioned in my comment 2, in the first version of the manuscript, it was difficult for me to understand and led to misunderstandings. But in the revised version, their intention to use the "frequency uncertainty model" became clear. The fitting error is the dominant uncertainty in the region of poor SNR ($SNR < 50$), and the baseline drift limits the uncertainty in the region of good SNR ($SNR > 50$). It would be an appropriate method to evaluate the uncertainty of the center frequency for these measurements. Other responses, including the responses to questions from other reviewers, are also clear and convincing. I believe that the revised manuscript is well written and ready for publication. I can recommend for publication in Nature communication.

#####

Reviewer #3 (Remarks to the Author):

The manuscript has been adequately revised and can be accepted for publication in NC.

#####